# Harnessing calcineurin-FK506-FKBP12 crystal structures from invasive fungal pathogens to develop antifungal agents

Praveen R. Juvvadi[1], David Fox III[2,3,4], Benjamin G. Bobay[5,6,7], Michael J. Hoy[8], Sophie M.C. Gobeil [6,7], Ronald A. Venters[5,6,7], Zanetta Chang[8], Jackie J. Lin[8], Anna Floyd Averette[8], D. Christopher Cole[1], Blake C. Barrington[1], Joshua D. Wheaton [9], Maria Ciofani [9], Michael Trzoss[10], Xiaoming Li[10,11], Soo Chan Lee [12], Ying-Lien Chen [13], Mitchell Mutz[10,14], Leonard D. Spicer[5,6,7], Maria A. Schumacher[6], Joseph Heitman[8] & William J. Steinbach[1,8]

Calcineurin is important for fungal virulence and a potential antifungal target, but compounds targeting calcineurin, such as FK506, are immunosuppressive. Here we report the crystal structures of calcineurin catalytic (CnA) and regulatory (CnB) subunits complexed with FK506 and the FK506-binding protein (FKBP12) from human fungal pathogens (*Aspergillus fumigatus*, *Candida albicans*, *Cryptococcus neoformans* and *Coccidioides immitis*). Fungal calcineurin complexes are similar to the mammalian complex, but comparison of fungal and human FKBP12 (hFKBP12) reveals conformational differences in the 40s and 80s loops. NMR analysis, molecular dynamic simulations, and mutations of the *A. fumigatus* CnA/CnB-FK506-FKBP12-complex identify a Phe88 residue, not conserved in hFKBP12, as critical for binding and inhibition of fungal calcineurin. These differences enable us to develop a less immunosuppressive FK506 analog, APX879, with an acetohydrazine substitution of the C22-carbonyl of FK506. APX879 exhibits reduced immunosuppressive activity and retains broad-spectrum antifungal activity and efficacy in a murine model of invasive fungal infection.

[1] Division of Pediatric Infectious Diseases, Department of Pediatrics, Duke University Medical Center, Durham, NC 27710, USA. [2] Beryllium Discovery Corp., 7869 NE Day Road West, Bainbridge Island, WA 98110, USA. [3] UCB Pharma., 7869 NE Day Road West, Bainbridge Island, WA 98110, USA. [4] Seattle Structural Genomics Center for Infectious Disease (SSGCID), Seattle, WA, USA. [5] Duke University NMR Center, Duke University Medical Center, Durham, NC 27710, USA. [6] Department of Biochemistry, Duke University, Durham, NC 27710, USA. [7] Department of Radiology, Duke University, Durham, NC 27710, USA. [8] Department of Molecular Genetics and Microbiology, Duke University Medical Center, Durham, NC 27710, USA. [9] Department of Immunology, Duke University Medical Center, Durham, NC 27710, USA. [10] Amplyx Pharmaceuticals, 3210 Merryfield Row, San Diego, CA 92121, USA. [11] Forge Therapeutics, Inc., 10578 Science Center Drive, San Diego, CA 92121, USA. [12] South Texas Center for Emerging Infectious Diseases, Department of Biology, The University of Texas at San Antonio, San Antonio, TX 78249, USA. [13] Department of Plant Pathology and Microbiology, National Taiwan University, Taipei 10617, Taiwan. [14] Genentech Inc., 1 DNA Way, San Francisco, CA 94080, USA. Correspondence and requests for materials should be addressed to P.R.J. (email: praveen.juvvadi@duke.edu) or to W.J.S. (email: bill.steinbach@duke.edu)

Calcineurin (CN) is a $Ca^{2+}$/calmodulin (CaM)-dependent, serine, threonine-specific protein phosphatase that controls T-cell activation and is critical for numerous cellular functions[1]. The CN complex is composed of a catalytic (CnA) and regulatory (CnB) subunit that binds $Ca^{2+}$, and is activated by $Ca^{2+}$–CaM[2]. CN inhibitors tacrolimus (FK506) and cyclosporine A (CsA) are immunosuppressive agents that have revolutionized transplantation[3]. CN activation triggers nuclear translocation of dephosphorylated NFAT transcription factors, and CN inhibition by FK506 and CsA blocks NFAT activity and T-cell proliferation, preventing graft rejection[4]. To function, these immunosuppressants must bind to their respective immunophilins, FK506-binding protein (FKBP12) and cyclophilin A (CypA)[5]. The immunophilin–immunosuppressant complexes (FK506-FKBP12; CsA-CypA) inhibit CN by binding to distinct overlapping regions in a hydrophobic groove formed by an extended α-helical region of CnA (CnB-Binding Helix, i.e. BBH) and the associated CnB subunit[6]. Structures of mammalian CN revealed important residues in CN binding and inhibition[6–8]. FK506 and CsA do not target the active site on CN, but rather sterically block access of phosphatase substrates.

The major human fungal pathogens, *Aspergillus fumigatus*, *Candida albicans*, and *Cryptococcus neoformans*, are responsible for over 1.6 million annual cases of invasive fungal infections worldwide, with mortality rates of up to 95%[9]. Engineering effective antifungals is challenging because fungi are eukaryotes, and require selective targeting to prevent harming the human host. CN is critical for fungi and orchestrates growth, stress responses, drug resistance, and virulence[10]. While these key functions make CN an attractive antifungal target[11–13], the host immunosuppressive effects of current CN inhibitors make it difficult to exploit their antifungal potential due to CN conservation between mammals and fungi. Despite CN conservation, interrogation of protein–protein and protein–ligand interactions resolved at the molecular level would facilitate the design of novel antifungal therapeutics.

Here we report the crystal structures of fungal CN ternary complexes with FKBP12-FK506 from highly clinically relevant human fungal pathogens: *A. fumigatus*, *C. albicans*, *C. neoformans*, and *Coccidioides immitis*, reflecting the common invasive fungal pathogens, including a mold, yeasts, and a dimorphic endemic fungus. Despite sequence variations between the mammalian and fungal counterparts of CN and FKBP12, the X-ray characterized ternary complexes reveal high structural homology. The interface between CN and FKBP12-FK506 complexes is conserved across fungi, but differences are notable in the binding surfaces of FKBP12 and CnA in fungi versus mammalian proteins. X-ray structures, NMR-based inhibitor binding studies, site-directed mutagenesis of the *A. fumigatus* CN–FK506-FKBP12 complex, and molecular dynamic (MD) simulations reveal a key residue, Phe88, which is not conserved in the mammalian FKBP12 (His88), as essential for binding and inhibiting fungal CN. This enabled the rational design and development of a less immunosuppressive FK506 analog, APX879, with broad-spectrum in vitro antifungal activity and efficacy in a cryptococcal infection model. This study broadens translational insight into the development of novel non-immunosuppressive CN inhibitors for effective antifungal targeting.

## Results

### Fungal CN complexes show distinct FKBP12-FK506 contacts.

Crystal structures of CN–FK506-FKBP12 ternary complexes from four fungi were determined by molecular replacement to resolutions between 1.85 and 3.30 Å (Table 1). Each complex has three components: (1) the CnA subunit with catalytic domain and the BBH, (2) the CnB subunit, with four EF hands bound to $Ca^{2+}$ and wrapping around the BBH, and (3) FKBP12 bound to FK506, interfacing with CnA/CnB (Fig. 1a). Mammalian and fungal CnAs share 62–73% sequence identity in their catalytic domains (Supplementary Table 1). A flexible hinge in CnA separates the catalytic domain from the BBH, which forms the composite surface for FKBP12-FK506-complex binding. Fungal CnB structures are also conserved, with two well-resolved C-terminal EF-hands close to the FKBP12-FK506 binding surface and the two distal N-terminal EF-hands showing varying degrees of disorder, suggesting interaction with FKBP12-FK506 may stabilize the C-terminal EF-hands. Structural superimpositions showed that the fungal CN–FK506-FKBP12 complexes are remarkably similar to mammalian complexes (bovine PDB: 1TCO and human PDB: 4OR9) with Cα root mean square deviations (RMSDs) between 0.3 and 0.4 Å for the CnA catalytic domains (44–54% sequence identity) and 0.6–0.7 Å for the CnB + BBH region (52–58% sequence identity) (Fig. 1b). Overlay of the mammalian FKBP12-FK506 (42–60% sequence identity) with the fungal FKBP12-FK506 complex resulted in Cα RMSDs of 0.4–0.7 Å.

In human FKBP12 (hFKBP12), the 40s and 80s loop residues serve as the effector face for interaction with the CN-complex (Fig. 1c). Previous work showed reduced affinity of hFKBP12 for CN with mutations in the 40s and 80s loops at residues Asp38, Arg43, His88, and Ile91[14]. While the CnA and CnB interacting residues within 5–8 Å of FK506 in human and fungi are conserved, fungal FKBP12s differed in 4 of the 17 interacting residues, as well as other residues surrounding the 40s and 80s loops (Fig. 1c). Comparison of fungal FKBP12s with hFKBP12 highlighted conformational differences at Lys48-Met50, Glu55, and His88 (Gln48-Gln50, Arg55, and Phe88 in *A. fumigatus*) (Fig. 1d; Supplementary Fig. 1a–d).

To investigate if these structural differences between the fungal and human counterparts are important for FK506-binding, NMR-based ligand binding experiments were performed. For NMR analysis, MD simulations, and genetic characterizations, we utilized *A. fumigatus* FKBP12 (AfFKBP12) as a model representing all fungal FKBP12s due to structural and sequence similarities. NMR analyses showed large chemical shift factors for Gly52, Glu55, Glu62, and Ala65 residues in hFKBP12 not observed in AfFKBP12 upon FK506 binding (Fig. 2a). Conversely, AfFKBP12 residues Ile25, His26, Tyr27, Val102, and Glu103 showed large chemical shift factors upon FK506 binding not observed in hFKBP12. Among these residues, only Ile25, Tyr27, and Glu55 (Arg55 in AfFKBP12) are close to FK506 (≤8 Å distance). Residues Thr49, Phe88, and Val91 did not show any significant changes. Importantly, these residues are not conserved between mammalian and fungal species and, based on the crystal structures, are likely more important for binding to the CnA/CnB interface.

### Mutations in *A. fumigatus* FKBP12 alter FK506 susceptibility.

To investigate the physiological relevance of the chemical shifts and sequence variations for fungal FKBP12-FK506-complex binding to CN and in vivo inhibition, mutations were introduced at the non-conserved residues surrounding the 40s loop and near those showing differential chemical shifts upon FK506 binding. The AfFKBP12 residues Phe22, Gln50, and Arg55 were mutated to the corresponding hFKBP12 residues Thr22, Met50, and Glu55 (F22T, Q50M, and R55E) (Supplementary Fig. 1e, f). Phe22 residue was mutated due to its structural proximity to Gln50, and to probe for second shell effects. The AfFKBP12 Phe88 residue in the 80s loop was also mutated to the corresponding hFKBP12 residue His88 (F88H) based on its proximity to FK506 (Supplementary Fig. 1d) to assess its contribution to

**Table 1 Crystallography and structure refinement statistics of calcineurin ternary complexes from pathogenic fungi**

| PDB ID | 6TZ6 | 6TZ7 | 5B8I | 6TZ8 |
|---|---|---|---|---|
| **Species** | *C. albicans* | *A. fumigatus* | *C. immitis* | *C. neoformans* |
| *Data collection* | | | | |
| Space group | $P2_12_12_1$ | $P2_1$ | $P2_12_12$ | $P2_12_12_1$ |
| Cell dimensions | | | | |
| $a, b, c$ (Å) | 62.47, 142.85, 175.61 | 59.24, 94.46, 69.83 | 94.67, 154.63, 64.75 | 118.72, 120.70, 134.96 |
| $\alpha, \beta, \gamma$ (°) | 90.00, 90.00, 90.00 | 90.00, 109.28, 90.00 | 90.00, 90.00, 90.00 | 90.00, 90.0, 90.00 |
| Resolution (Å) | 2.55 (2.62–2.55) | 2.50 (2.56–2.50) | 1.85 (1.90–1.85) | 3.30 (3.39–3.30) |
| $CC(1/2)$ | 99.8 (82.6) | 99.6 (72.8) | 99.8 (92.3) | 99.2 (91.8) |
| $CC^*$ | 99.9 (95.3) | 99.9 (91.7) | 99.9 (98.2) | 99.8 (98.0) |
| $I/\sigma(I)$ | 16.87 (3.37) | 12.00 (2.09) | 15.70 (4.20) | 8.62 (3.34) |
| Completeness (%) | 98.4 (99.2) | 99.7 (99.9) | 100.0 (100.0) | 99.9 (99.9) |
| Redundancy | 4.4 (4.5) | 3.8 (3.9) | 7.4 (7.5) | 6.2 (6.3) |
| *Refinement* | | | | |
| Resolution (Å) | 2.55 | 2.5 | 1.85 | 3.3 |
| No. of reflections | 51,429 | 25,206 | 81,827 | 29,795 |
| $R_{work}/R_{free}$ overall | 18.2/22.9 | 20.0/23.2 | 14.5/17.2 | 19.9/24.3 |
| Molprobity | | | | |
| Ramachandran | | | | |
| Favored (%) | 96.43 | 96.07 | 96.89 | 94.84 |
| Outliers (%) | 0.17 | 0.00 | 0.16 | 0.31 |
| Rotamers | | | | |
| Favored (%) | 92.62 | 96.82 | 96.32 | 90.04 |
| Outliers (%) | 2.81 | 0.64 | 1.23 | 1.93 |
| MolProbity score | 1.83 (98th) | 1.28 (100th) | 1.35 (98th) | 1.78 (100th) |
| No. of atoms | | | | |
| Protein | 9429 | 4643 | 5260 | 10,132 |
| CnA | 5749 | 2879 | 2974 | 6028 |
| CnB | 2046 | 875 | 1374 | 2510 |
| FKBP12 | 1634 | 889 | 912 | 1594 |
| Ligand | 148 | 75 | 126 | 141 |
| FK506 | 114 | 57 | 57 | 114 |
| Water | 281 | 103 | 687 | 9 |
| ADP (Å$^2$) | | | | |
| Protein | 52.66 | 47.01 | 25.68 | 40.20 |
| CnA | 42.31 | 43.43 | 20.04 | 36.21 |
| CnB | 70.32 | 62.50 | 29.89 | 49.00 |
| FKBP12 | 66.97 | 43.38 | 37.76 | 41.42 |
| Ligand | 47.41 | 41.17 | 27.90 | 32.89 |
| FK506 | 43.20 | 36.08 | 19.71 | 27.03 |
| Water | 37.95 | 39.65 | 37.62 | 20.99 |
| r.m.s. deviations | | | | |
| Bond lengths (Å) | 0.008 | 0.002 | 0.02 | 0.005 |
| Bond angles (°) | 0.916 | 0.597 | 1.641 | 1.03 |

FK506 stabilization. Phenotypic analyses of *A. fumigatus* strains revealed that the F22T and Q50M mutations induced partial resistance to FK506, while the R55E mutant remained sensitive (Fig. 2b). Strikingly, the F88H mutation caused enhanced resistance to FK506, suggesting reduced or complete loss in affinity of FKBP12 to FK506 and/or the CN-complex (Fig. 2b). Because *Affkbp12* gene deletion resulted in FK506 resistance[15], we confirmed the stability of mutated AfFKBP12 proteins to rule out any degradation as a consequence of the mutations (Supplementary Fig. 1g).

Previous studies showed that the CN-complex localized at the hyphal septum[16], and the AfFKBP12-GFP fusion protein bound to CN at the septum in the presence of FK506[15]. Under normal growth conditions, AfFKBP12 proteins localized to the cytoplasm and nuclei (Fig. 2c). Following FK506 treatment, AfFKBP12-F22T and AfFKBP12-R55E proteins localized at the septum, indicating their binding to CN at that site (Fig. 2c). In contrast, the AfFKBP12-F88H protein did not localize at the septum at growth inhibitory concentrations of FK506 (0.1 µg/ml) (Fig. 2c), indicating a loss in CN binding, and correlating with the observed complete resistance of the AfFKBP12-F88H strain to FK506 at

0.1 µg/ml (Fig. 2b). Only at higher concentrations of FK506 (0.5 and 1 µg/ml) was septal localization (17% and 29%, respectively) of AfFKBP12-F88H observed (Supplementary Fig. 1h), suggesting higher drug concentrations might enhance the formation of the FKBP12–FK506-complex or enhance CN binding in vivo. Intriguingly, the partially resistant AfFKBP12-F22T and AfFKBP12-Q50M strains showed interaction of the respective AfFKBP12 mutant proteins to CN in vivo, suggesting that in contrast to Phe88, the Phe22 and Gln50 residues may not be critical for or directly involved in CN–FKBP12–FK506-complex formation.

The structural basis for the observed FK506 resistance of *Affkbp12* mutants was evaluated using *A. fumigatus* and mammalian CN-complex crystal structures, as well as MD-simulated structures of unbound WT-AfFKBP12 and the AfFKBP12-F22T, AfFKBP12-Q50M, AfFKBP12-R55E, and AfFKBP12-F88H mutants. Based on the crystal structures alone we would not expect mutation of Phe22, Gln50, or Arg55 to impact FK506 resistance as they either reside distal (>10 Å) or make no conserved contacts with CN or FK506 (Fig. 1d, panels 1 and 3). However, MD simulation of all the unbound mutants

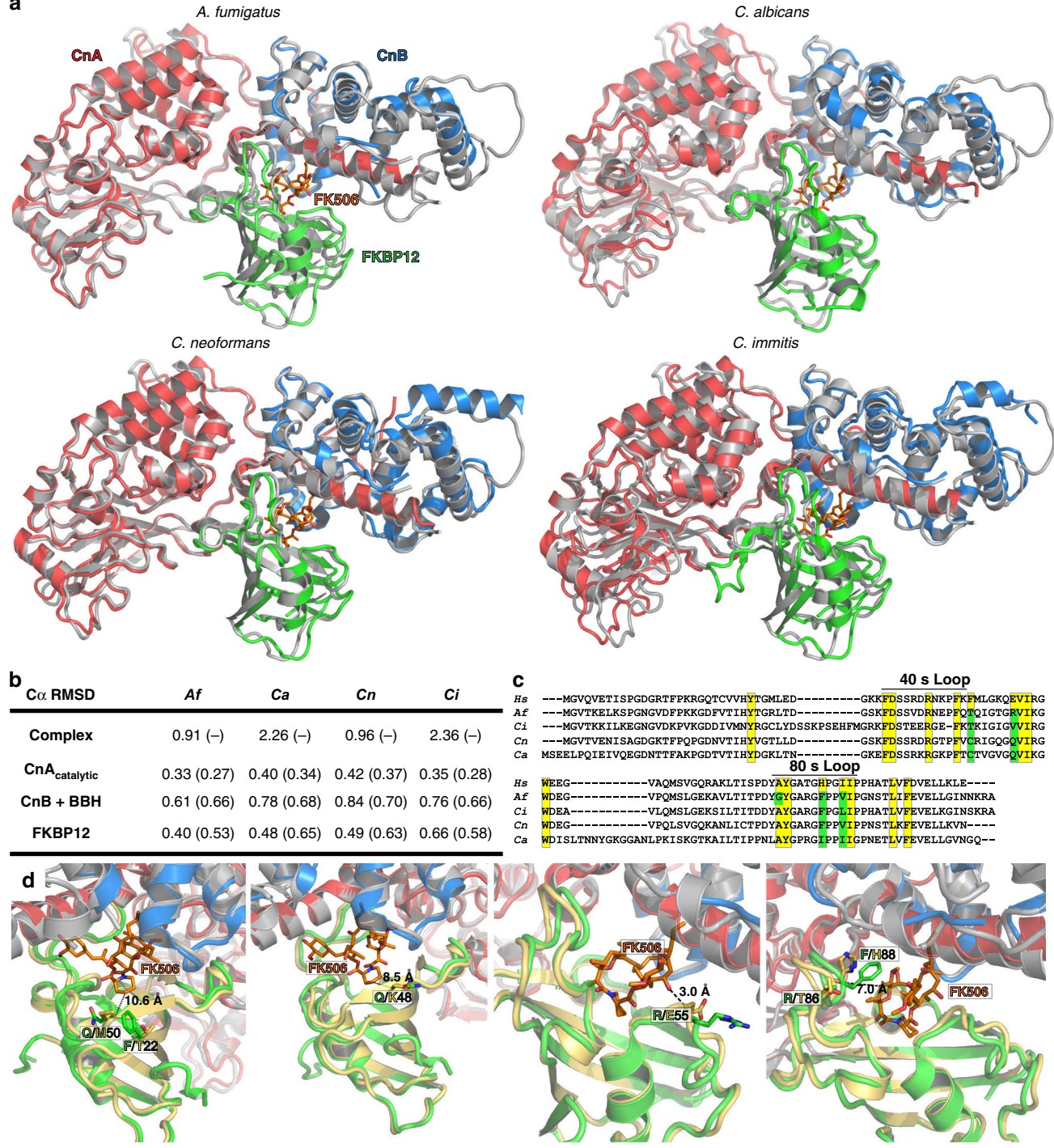

**Fig. 1** Crystal structures of fungal CN–FK506-FKBP12 complexes. **a** Overlay of fungal CNs with the bovine complex (bovine: PDB 1TCO gray; fungal: CnA red, CnB blue, FKBP12 green and FK506 in orange). **b** RMSD of fungal CN complexes to the bovine complex and isolated components with *Homo sapiens* CN complex (CnA/CnB; PDB 4OR9) and *H. sapiens* FKBP12 (PDB 2PPN) in parentheses. **c** Alignment of human and fungal FKBP12s. Residues within 5 Å of FK506 are highlighted yellow. Non-conserved residues in fungi are highlighted green. **d** Structural comparison of variable residues in *A. fumigatus* (colored as in **a**; PDB 6TZ7) and *H. sapiens* FKBP12 (colored yellow; PDB 2PPN). Note that both AfFKBP12 F22 and Q50 reside distal (>10 Å) to the FKBP12-FK506-binding site on CnA or CnB (panel 1). The side-chains of AfFKBP12 R55 and hFKBP12 E55 are near the BBH and CnB, but face away from FK506 and do not make conserved contacts to CnA/CnB other than a distal main-chain carbonyl interaction with FK506 (panel 3). See also Supplementary Fig. 1 and Supplementary Tables 1 and 2

revealed Cα RMSD variations compared to the WT protein in residues 41–45, 52–54, and 86–91 (or there within), suggesting altered conformational space that may or may not be tolerated in forming complexes with CN. While the CnA-binding surface in the F22T and Q50M mutations remained unchanged, the R55E

mutation showed an altered CnA-binding surface. An altered FK506-binding cavity was noted for the F22T and F88H mutants (in size and polarity), while Q50M and R55E have a near WT FK506-binding cavity (Fig. 2d–g). Despite altered FK506-binding cavities, these mutants may still bind FK506, albeit with lower

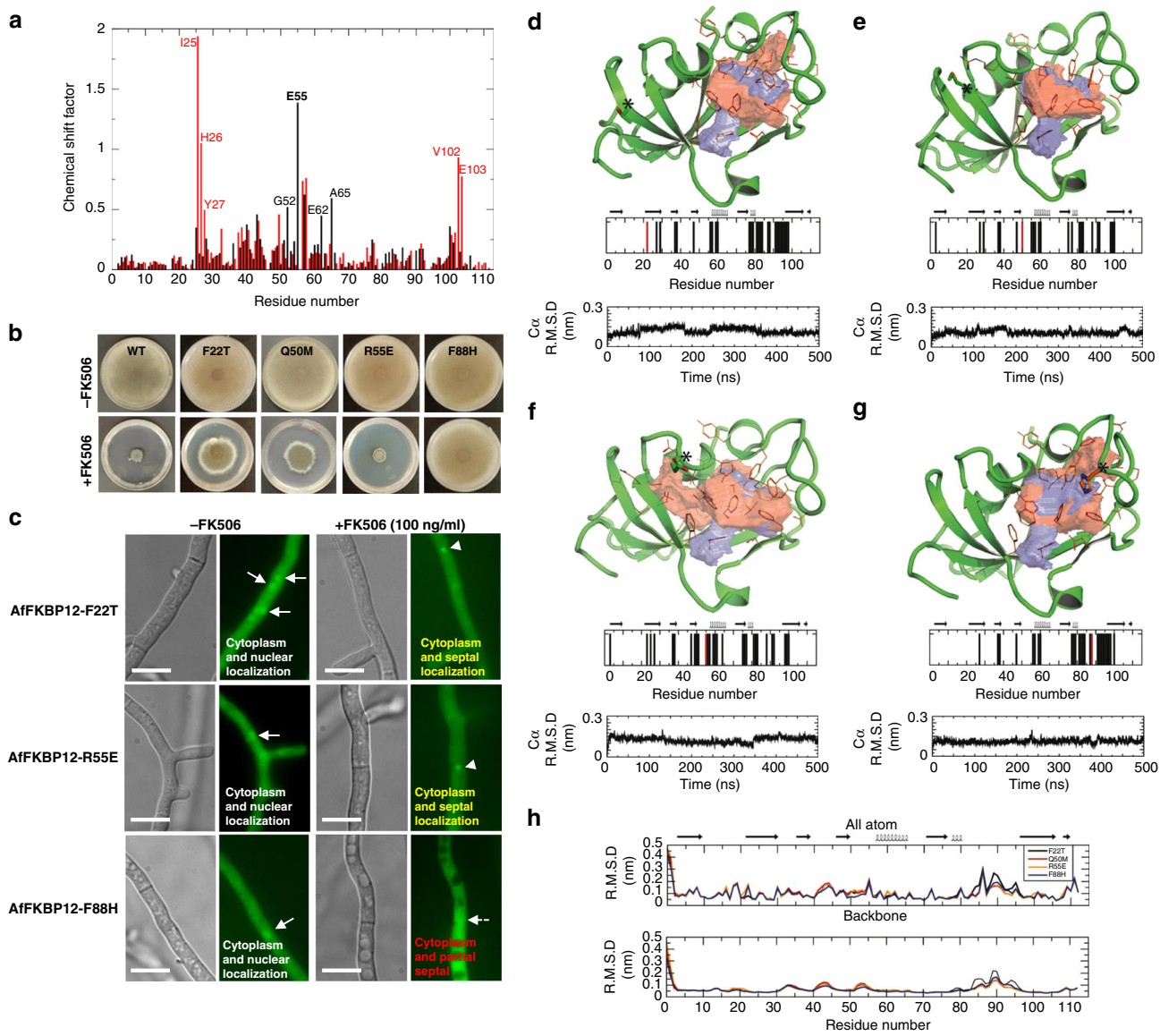

**Fig. 2** Structural implication of AfFKBP12 mutations. **a** NMR analysis showing chemical shifts induced by FK506 binding to *A. fumigatus* (red) and human (black) FKBP12. **b** Growth of the *Affkbp12* mutants in the absence and presence of FK506 for 5 days at 37 °C. **c** Microscopic localization AfFKBP12 mutated proteins in vivo in the absence or presence of FK506. Arrows show nuclear localization of AfFKBP12. Arrowheads indicate binding of AfFKBP12 to CN at the hyphal septum (**d** F22T, **e** Q50M, **f** R55E, **g** F88H). The FK506-binding pocket of WT (blue surface) and the pocket resulting from the mutation (red surface) are shown. Mutation site is denoted with an "*" and shown in stick format (red bar in bar graph), while residues comprising the binding pocket are shown in orange lines (black bars in bar graph). The stability and quality of the simulation is shown in the Cα RMSD plot. **h** The all atom (top) and backbone (bottom) RMSD denoting the structural variation due to the mutation compared to the WT structure (5HWB). See also Supplementary Figs. 1 and 3

affinity. They may also have structural defects, such as increased RMSD in the 80s loop (Fig. 2h), which could disrupt CN contacts and result in decreased FK506 sensitivity.

Interestingly, only the F88H mutation led to significantly increased FK506 resistance but maintained its affinity/interaction for FK506 as observed in NMR titrations (Supplementary Fig. 2a, b). MD simulations of F88H mutation (which resides within 5 Å of FK506 and CnA/CnB interface) showed changes in the CnA-binding surface, apolar/polar composition of the FK506-binding cavity, and large Cα RMSD variations in the 80s loop compared to WT-AfFKBP12 (Fig. 2g, h). X-ray structures of AfFKBP12 and mammalian FKBP12 revealed conformational differences in Phe88/His88 positioning and the associated 80s loop. In AfFKBP12, the aromatic side-chain of Phe88 is partially buried at the FK506, FKBP12, and BBH interface (SASA = 43.8 Å², 34%),

providing important hydrophobic contacts. In the mammalian FKBP12, His88, while also partially buried at the same interface (Solvent-accessible surface area (SASA) = 52.1 Å², 44%), mediates a hydrogen bond (H-bond) to a conserved water coordinated between FK506 and BBH not observed in the AfFKBP12-complex (Supplementary Fig. 2c). Partly due to these differences, the mammalian 80s loop packs tighter against the BBH and FK506. Given these differences, MD simulations were used to investigate formation of the CnA/CnB–FK506–AfFKBP12–F88H-complex to visualize the structural basis for FK506 resistance (Supplementary Fig. 3). Scoring the binding of AfFKBP12–FK506 to CN for both the WT and F88H mutant for each structure over the last 100 ns of simulation resulted in an average score of −7.5 and −9.1 kcal/mol, 3.3 μM versus 200 nM, respectively. Analysis of the X-ray structures and MD simulation suggest the inability of the

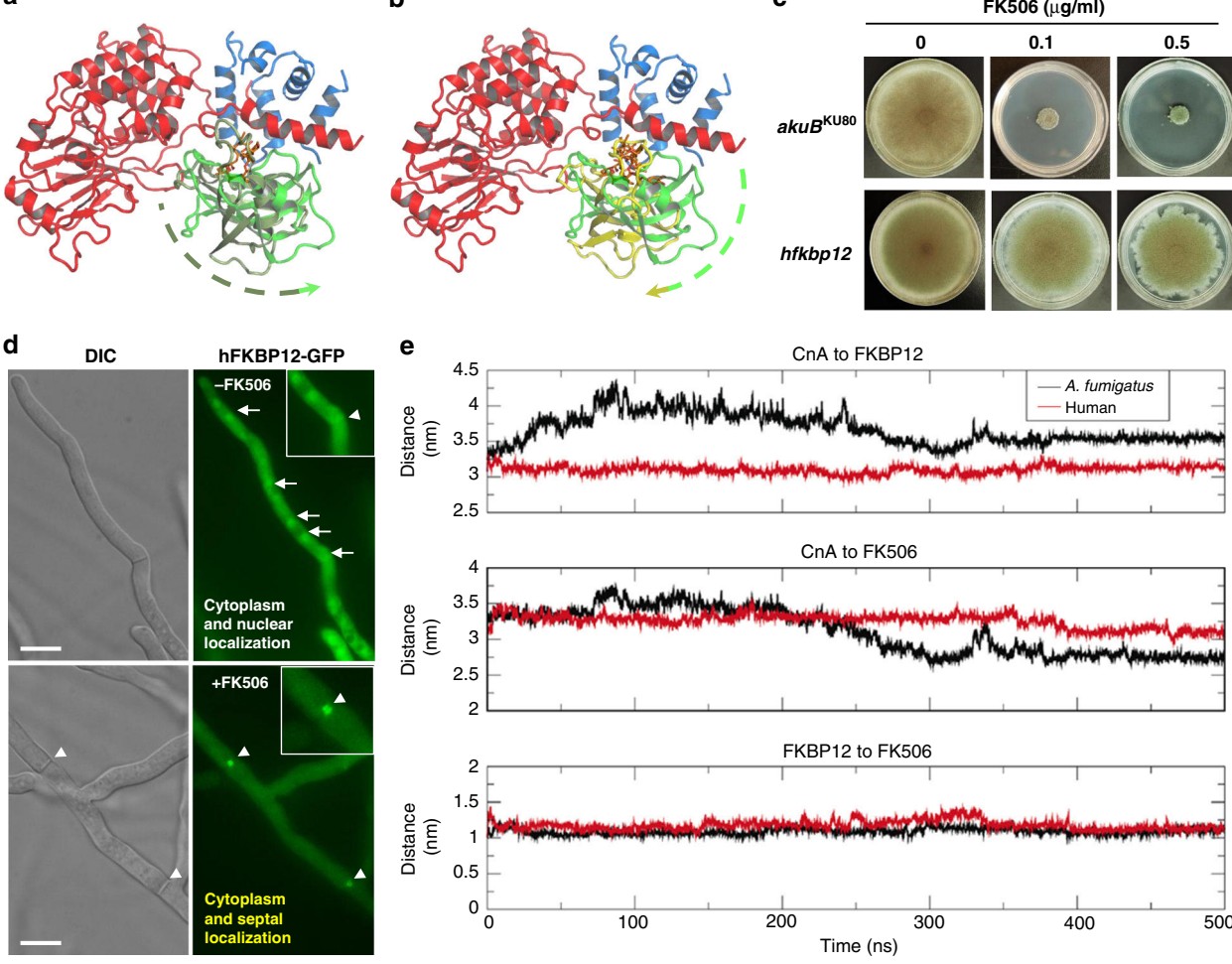

**Fig. 3** X-ray and MD-simulated structures of *A. fumigatus* and human CN-complex. **a** Conformational rotation of FKBP12 about the FK506-binding pocket comparing the X-ray characterized and MD characterized CnA (red)/CnB(blue)–FKBP12 [olive (X-ray) and green (MD simulated)]–FK506 complex. **b** Conformational rotation of FKBP12 about the FK506-binding pocket comparing the MD characterized *A. fumigatus* complex and the MD characterized human CnA (red)/CnB(blue)–FKBP12 [green (*A. fumigatus*) and yellow (human)]–FK506 complex. Rotation in **a** and **b** are depicted by dashed arrow lines color coded based on the FKBP12 coloring in each panel. **c** Growth of the *A. fumigatus* strain expressing codon-optimized hFKBP12 in the absence and presence of FK506 for 5 days at 37 °C. **d** Microscopic localization hFKBP12-GFP in *A. fumigatus* in the absence or presence of FK506. Arrows show nuclear localization of hFKBP12. Arrowheads indicate the binding of hFKBP12 to CN at the hyphal septum. **e** Center of mass (COM) distances between different chains convey the stability of the complexes over the last 100 ns; top—CnA to FKBP12, middle—CnA to FK506, and bottom—FKBP12 to FK506. See also Supplementary Figs. 4 and 5

AfFKBP12–F88H–FK506-complex to establish significant interaction with CN is due to the reduced affinity to CnA/CnB and not to perturbed FK506 binding. This is consistent with the observed interaction of AfFKBP12–F88H–FK506 in the NMR chemical shifts.

**AfFKBP12 and hFKBP12 differ in binding interface with CN.** We hypothesized that NMR chemical shifts indicate differential interaction between FKBP12–FK506 and fungal CN-complex compared to the human complex. Although significant similarities exist between the mammalian and *A. fumigatus* complexes, the binding surfaces between FKBP12 and CnA/CnB exhibit notable differences (Supplementary Table 2, Supplementary Fig. 4a–c). MD simulations of the *A. fumigatus* and human CN–FKBP12–FK506 ternary complexes revealed stability in the last 100 ns of each simulation that were compared and characterized based on the average structures from a Cα RMSD clustering analysis (Supplementary Fig. 3). A counter-clockwise rotation of AfFKBP12 about the FK506-binding region in relation

to the CN-complex was observed when comparing the MD-simulated structure to the X-ray structure (Fig. 3a). This ~10° rotation of the center of mass (COM) of FKBP12 with respect to the hinge between the catalytic domain of CnA and BBH resulted in a larger contact surface area between FKBP12 and CnA in comparison to the X-ray structure, while the individual domain RMSDs remained similar in conformation (<0.7 Å). In fact, the essential residues for AfFKBP12 and CnA-complex formation in the X-ray structure were also observed throughout the MD simulation. MD simulations also revealed a clockwise rotation (26.5° rotation between the COM of FKBP12 and the hinge between the catalytic domain of CnA and BBH) of FKBP12 about the FK506-binding region in relation to the CN-complex when comparing human and *A. fumigatus* further lending support to binding surface differences between them (Fig. 3b).

**hFKBP12 does not complement AfFKBP12 function.** Due to the binding interface differences between *A. fumigatus* and human FKBP12 and CN-complex, as well as the AfFKBP12–F88H

mutation conferring FK506 resistance, we next assessed if hFKBP12 complemented AfFKBP12 function. A codon-optimized *hfkbp12* cDNA allele for *A. fumigatus* was expressed from the native locus of *Affkbp12* (Supplementary Fig. 5a). The *A. fumigatus* strain expressing hFKBP12 did not exhibit any growth difference from the WT in the absence of FK506 (Fig. 3c and Supplementary Fig. 5b). Intriguingly, hFKBP12 did not complement AfFKBP12 function in the presence of FK506 despite the confirmed expression of hFKBP12 in *A. fumigatus* (Supplementary Fig. 5b). *A. fumigatus* expressing hFKBP12 did not show any growth defect with 0.1 µg/ml FK506, and 0.5 µg/ml FK506 caused only partial growth inhibition, suggesting that hFKBP12-FK506 is impaired in binding to *A. fumigatus* CN. In contrast, hFKBP12-GFP localized at the septum in the presence of 0.1 µg/ml FK506, indicative of its binding to CN (Fig. 3d) and suggesting that the complex forms in vivo, sufficient for colocalization, but insufficient for inhibition.

X-ray structures and MD simulations provided a structural basis for the observed phenotypes of the FKBP12 mutants. Mammalian FKBP12–FK506 aligned onto the *A. fumigatus* X-ray structure with a Cα RMSD of 0.4 Å, highlighting overall structural similarity. Despite the similarity, significant differences were observed in FKBP12 near FK506 and the BBH/CnB interface. These include differences throughout the loop between Phe49-Glu55 (hFKBP12), which appears to stem from the Phe → Thr (F49/T49) substitution between hFKBP12 and AfFKBP12, respectively (Supplementary Fig. 5c). Additionally, the 80s loop residues His88-Ile91 in hFKBP12 (Phe88-Val91 in AfFKBP12) contribute directly to FK506 binding and to the interface with BBH and CnA catalytic domain C-terminal residues. Key differences include substitution of the hydrophobic and aromatic side-chain of Phe88 with the hydrophilic and basic side-chain of His88, leading to displacement in the associated loop. Together, differences between species within the 40s and 80s loops impact the conformation of FKBP12 relative to FK506 and the BBH. Monitoring the COM distances of CnA to FK506 in MD simulation revealed that hFKBP12–FK506 migrates away from *A. fumigatus* CnA compared to WT–AfFKBP12–FK506, 3.2 nm versus 2.8 nm, respectively, while the FKBP12–FK506 interaction remains intact (Fig. 3e). These critical differences suggest that hFKBP12–FK506 binding to *A. fumigatus* CN is significantly weaker than native AfFKBP12–FK506 binding to *A. fumigatus* CN. MD simulation data suggest a 200 nM (−9.1 kcal/mol) versus 120 µM (−5.3 kcal/mol) affinity, which is supported by the differences observed in the X-ray structures.

**Mutations in hFKBP12 complement AfFKBP12 function**. To identify residues important for the lack of FK506 activity in *A. fumigatus* expressing hFKBP12, the non-conserved residues in the 40s and 80s loops of hFKBP12 were substituted for corresponding residues of AfFKBP12 (Fig. 4a; Supplementary Fig. 5d). While Met50 (M50Q) mutation induced partial sensitivity to FK506 (0.5 µg/ml), the Lys48 (K48Q) mutation showed increased sensitivity in comparison to the WT–hFKBP12 expressing strain. Double mutation of Met50 and Lys48 (K48Q-M50Q) yielded an additive effect with higher FK506 sensitivity. However, double mutation of Arg41 and Lys45 (R41V-K45E) or Lys53 and Gln54 (K53T-Q54G) did not alter FK506 susceptibility. While X-ray structures suggest that these residues mediate additional contacts between FKBP12 and either CnB or CnA distal from the BBH and the FK506-binding surface, MD simulations suggest that these residues make direct contact with the BBH through rotation of FKBP12. Interestingly, single mutations in the 80s loop at residues Thr86 (T86R) or His88 (H88F) conferred FK506 sensitivity (Fig. 4a). Double mutation of Thr86 (T86R) and His88 (H88F) led

to complete recovery of FK506 sensitivity. These results indicate that Phe88 is a key residue required for AfFKBP12–FK506 to interact with and inhibit *A. fumigatus* CN. These data further support our observation that the AfFKBP12–F88H mutation resulted in complete loss of FK506 sensitivity (Fig. 2b). Therefore, in the hybrid complex, burial of the hydrophobic side-chain of Phe88 appears better able to compensate for the differences in the 40s and 80s loops of the complex than the native hydrophilic His88 side-chain.

In addition to analysis of the hybrid model based on the crystal structures, structural basis for FK506 sensitivity in hFKBP12 mutants K48Q, M50Q, and H88F was investigated using MD simulations of unbound hFKBP12. These revealed Cα RMSD variations compared to the WT–hFKBP12 throughout residues 10–20, 32–35, 41–45, and the 80s loop, suggesting altered conformational space that may or may not be tolerated in forming complexes with CnA/CnB. While MD simulations of K48Q, M50Q, and K48Q–M50Q revealed a near WT FK506-binding cavity, the H88F mutation led to an increase in polar characteristics. This may be due to the buried Phe residue causing exposure of adjacent FK506-binding residues, such as Arg41, Arg43, Lys45, and/or Lys48 (Fig. 4b–f). Structural effects of the T86R mutation in hFKBP12 (as extrapolated from the simulation of hFKBP12–H88F–FK506 binding to *A. fumigatus* CN-complex) may be attributed to additional positive electrostatic surface character, similar and stronger compared to the H88F mutation (Fig. 4e).

We postulated that the increased FK506 susceptibility with these mutations in hFKBP12 might be due to more native-like AfFKBP12 interactions with either FK506 or with *A. fumigatus* CN-complex. However, NMR titrations revealed no significant variation between the binding of AfFKBP12–F88H and hFKBP12–H88F to FK506 (Supplementary Figs. 2a, b and 6a, b). This suggests that mutations in hFKBP12 influence the interaction of hFKBP12–FK506-complex to *A. fumigatus* CN, and not hFKBP12 binding to FK506. MD simulations of hFKBP12–H88F–FK506 and WT–AfFKBP12–FK506–CnA/CnB ternary complexes suggested recovery in similar binding affinity of hFKBP12–H88F–FK506 compared to WT–AfFKBP12–FK506 when binding to the *A. fumigatus* CN-complex, 330 nM (−8.8 kcal/mol) versus 200 nM (−9.1 kcal/mol), respectively, further supporting the induction of FK506 sensitivity in the hFKBP12–H88F mutant.

**Key FKBP12–FK506 contact residues in *A. fumigatus* CnA**. An earlier study showed that mutations in the *A. fumigatus* BBH residues Asn367, Trp374, or Ser375 (N367D, W374L, S375T) resulted in FK506 resistance[17]. In support of these, the crystal structure of *A. fumigatus* CnA shows significant interactions from these residues that are conserved in human CnA. Asn367 makes a H-bond with Pro90 in FKBP12, but this interaction is absent in the mammalian CN-complex due to the P90G substitution in FKBP12 and L186Q substitution in CnA (Supplementary Fig. 4c). Trp374 in the mammalian complex makes a bifurcated H-bond to FK506 O7 and O8, and has its side-chain completely buried when complexed with FKBP12–FK506. The W374L mutation would disrupt these key H-bonds with FK506 and result in a small cavity between the BBH and FK506. The Ser375 also makes a conserved bifurcated H-bond with CnB Tyr121 and CnA Val371. S375T mutation would add a methyl group within clashing distance of the FK506 alkyl arm at C40, leading to the disruption of BBH–FK506 interactions. MD simulations of native AfFKBP12–FK506 binding to *A. fumigatus* CnA/CnB also revealed that Trp374 and Ser375 within the BBH not only have significant contact with FK506, but also with AfFKBP12 Phe47,

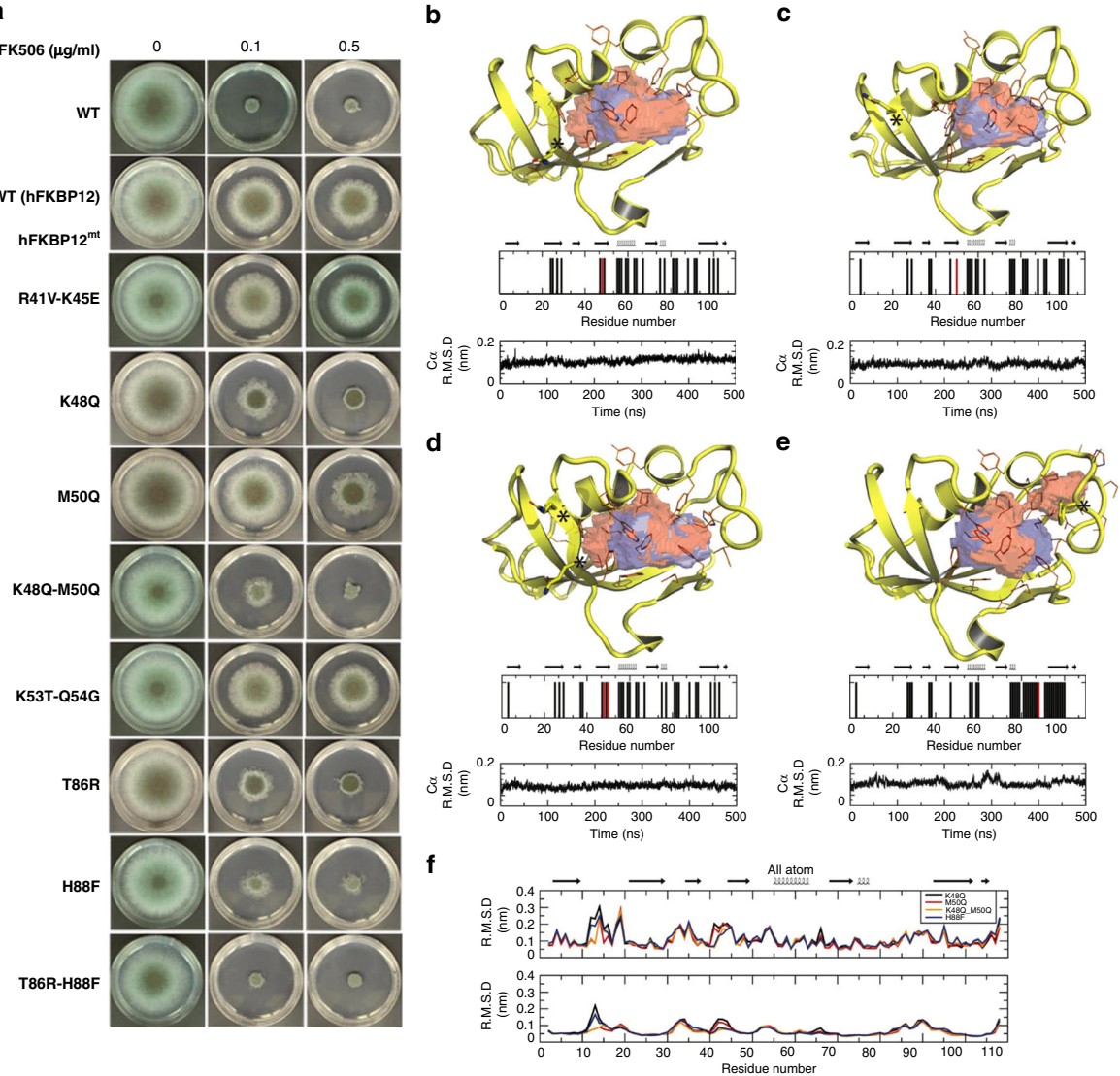

**Fig. 4** FK506 susceptibility and structural implication of hFKBP12 mutants. **a** Growth of the *hFKBP12* mutants in the absence and presence of FK506 for 3 days at 37 °C (**b** K48Q, **c** Q50M, **d** K48Q–Q50M, **e** H88F) The FK506-binding pocket of WT (blue surface) and the pocket resulting from the mutation (red surface) are shown. Mutation site is denoted with a "*" and shown in stick format (red bar in bar graph), while residues comprising the binding pocket are shown in orange lines (black bars in bar graph). Stability and quality of the simulation is shown in the Cα RMSD plot. **f** All atom (top) and backbone (bottom) RMSD denoting the structural variation due to the mutation compared to the WT structure (1FKF-monomer). See also Supplementary Figs. 6 and 11

Arg55, and Phe88 in the 40s and 80s loops (Fig. 5a). Of these AfFKBP12 residues in contact with the BBH, only the conserved Phe47 had a significant number of contacts with CnA Trp374 and Ser375. MD simulations also showed that Tyr363 and Met369 in the BBH have contacts with both FK506 and FKBP12 Arg43, Phe47, and Phe88. In the human complex, the FKBP12 residues that contact the BBH within CnA were Phe37, Asp38, Ser39, Asp42, Arg43, Lys45, His88, Pro89, Gly90, and Ile91 (Fig. 5b). In both *A. fumigatus* and human complex MD simulations, Arg43 makes the majority of the contacts to residues in the BBH (conserved residue in 40s loop of fungal FKBP12s), followed by residues within the 80s loop region.

To identify other key CnA residues important for complex formation with AfFKBP12–FK506, we generated a series of mutations in CnA-BBH at residues Tyr363, Trp364, Met369, Thr373, and Thr384 (Supplementary Fig. 7a). While Tyr363 (Y363A, Y363F) and M369A mutations induced partial FK506 resistance, simultaneous mutation of Tyr363 and Met369

(Y363A–M369A) led to increased FK506 resistance (Supplementary Fig. 7b). Mutations at other residues (Trp364; Thr373; Thr384) did not alter FK506 sensitivity, indicating the importance of Tyr363 and Met369 contacts with FK506 and the 40s and 80s loops in FKBP12. In the *A. fumigatus* crystal structure, Tyr363 and Met369 are located N-terminal to the BBH as it transitions from the catalytic domain and these are conserved within human CnA. The side-chain for Met369 inserts itself into the small hydrophobic pocket with Leu365, Thr373, and Trp374 of CnA and Phe88, Pro90, and Val91 of AfFKBP12, and packs against FK506. An M369A mutation would decrease the hydrophobic complementary surface area between CnA and FKBP12–FK506, potentially destabilizing the conformation of the 80s loop and the side-chain of Phe88, shown to be critical for FK506 resistance. The CnA Tyr363 forms a direct H-bond with AfFKBP12 Arg43, a key residue in stabilizing interactions with the acidic side-chain of Asp38, which directly contacts FK506 O4 and O5. Therefore, Tyr363 represents an important polar contact

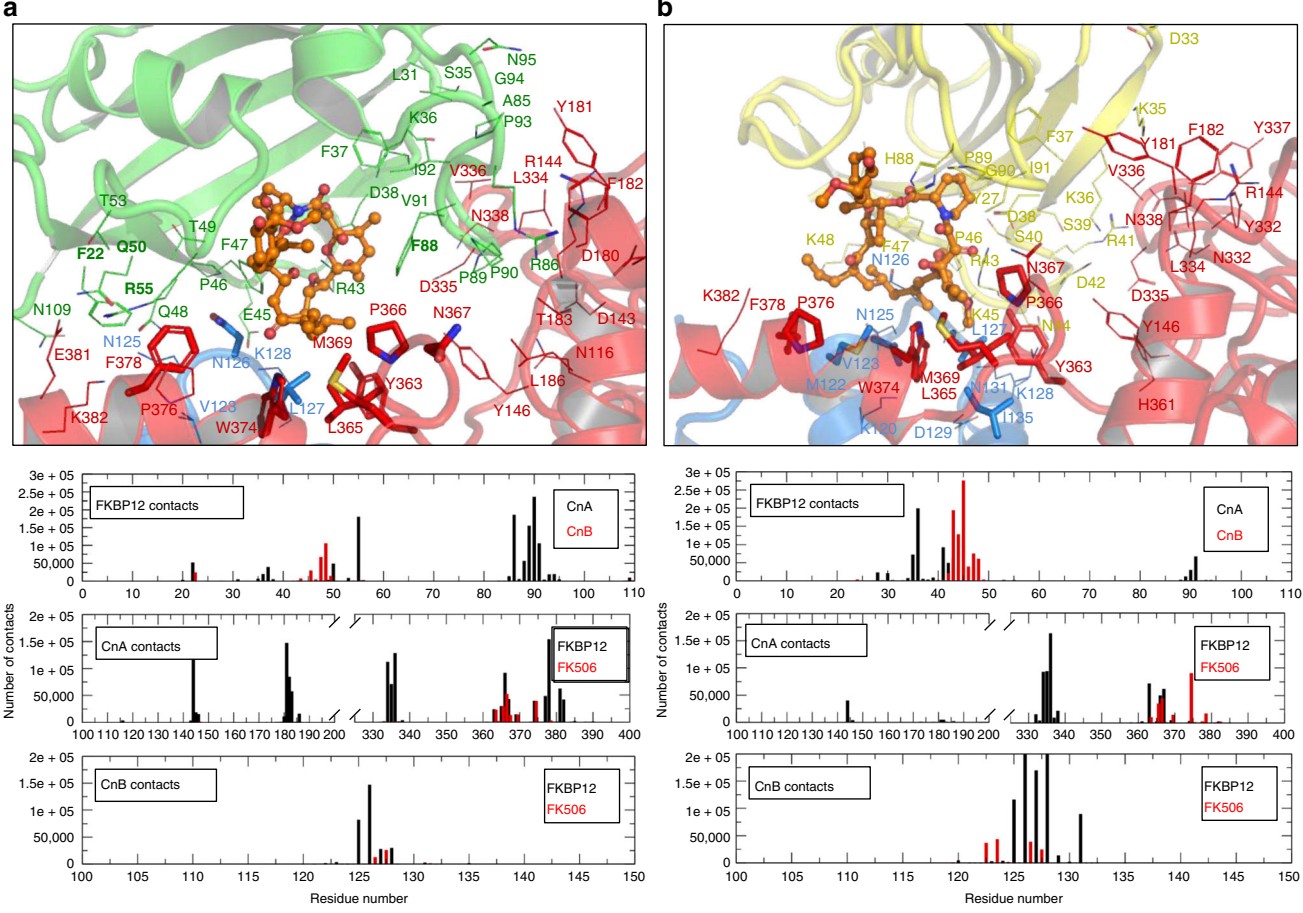

**Fig. 5** Binding of FKBP12–FK506 to *A. fumigatus* CN. **a** Cartoon representation of *A. fumigatus* WT AfFKBP12–FK506 complex binding to *A. fumigatus* CN (CnA/CnB) from the MD simulation. Cartoon representations colored as follows: CnA (red), CnB (blue), AfFKBP12 (green), and FK506 (orange sticks and spheres). **b** Cartoon representation of Human hFKBP12:FK506 complex binding to *A. fumigatus* CnA/CnB from the MD simulation. Cartoon representations colored as follows: CnA (red), CnB (blue), hFKBP12 (yellow), and FK506 (orange sticks and spheres). Residue labels are shown for clarity and colored to the same color as their protein. Residues (those with more than 2000 contacts in the last 100 ns of the simulation) depicted as lines are contacts between either FKBP12 and CnA or CnB, while sticks are contacts between FK506 and CnA or CnB. Valence states are shown for label clarity. Bar graphs denote all contacts within 6 Å during the last 100 ns of each simulation; top bar graph, FKBP12 contacts to CnA (black) and CnB (red); middle graph, CnA contacts to FKBP12 (black), and FK506 (red), slanted lines denote residues 201–324 to which there are no contacts in CnA; bottom graph, CnB contacts to FKBP12 (black), and FK506 (red). See also Supplementary Figs. 7 and 8 and Supplementary Table 2

site between CnA and FKBP12–FK506, which is disrupted with either a Tyr to Ala or a Tyr to Phe mutation.

Based on the CnA/CnB–FK506–FKBP12 structure, the CnA loop 7 (residues 328–339; Supplementary Fig. 7) is proximal to the FKBP12–FK506 complex, suggesting involvement of loop 7 residues in FKBP12–FK506 interactions[7]. In accord with the X-ray structure, MD simulations also revealed that the loop 7 of CnA only interacts with AfFKBP12, and has no direct contact with FK506. The 11 residues in AfFKBP12 contacting CnA loop 7 included Leu31, Ser35–Asp38, Arg43, and Pro89–Pro93 (Fig. 5). Among these residues, Lys36, Phe37, Pro90, and Val91 have significant interactions with loop 7. While only three out of these 11 residues (Ser35, Pro90, and Val91) are not conserved in AfFKBP12, our earlier mutation of the Pro90 (P90G) and Val91 (V91C) residues demonstrated increased resistance to FK506[18], further lending support to their significance in making contacts with the CnA loop 7. AfFKBP12 40s loop residues Asp38–Asp42 primarily contact CnA residues Asn332, Asp335, and Val336, while in the human complex these residues largely contact residues Tyr146, Asn332, Asp335, Val336, Asn338, and His361 (Fig. 5). However, none of the mutations introduced in the important loop 7 residues (Leu334, Asp335, Val336, and Tyr337)

induced FK506 resistance, indicating a major role for the 80s loop residues in FKBP12 for establishing contact with the CnA BBH and the CnA loop 7 region. Taken together, the AfFKBP12 80s loop residues (Arg86, Phe88, Pro90, and Val91), which are not conserved in hFKBP12, are important and required for the inhibition of *A. fumigatus* CN.

**The FK506 analog APX879 mimics FK506 inhibition of CN.** We next sought to capitalize on these structural insights to design FK506 analogs that exhibit reduced or abrogated mammalian immunosuppressive activity yet retain antifungal activity. The two non-conserved residues in fungal FKBP12, Phe88, and Val91, at the interface between FKBP12–FK506 and CnA/CnB were exploited for this purpose. Based on our structural model, the proximity of these residues to the C22-carbonyl of FK506 in the bound conformation was an ideal starting point for semi-synthetic modifications of the complex macrolide. We hypothesized that small derivatives of the C22-carbonyl group would sterically clash to a greater extent with His88 of the mammalian complex when compared to Phe88 in the fungal complex. To test this, a series of oximes and alkyl- and hydrazide hydrazones were

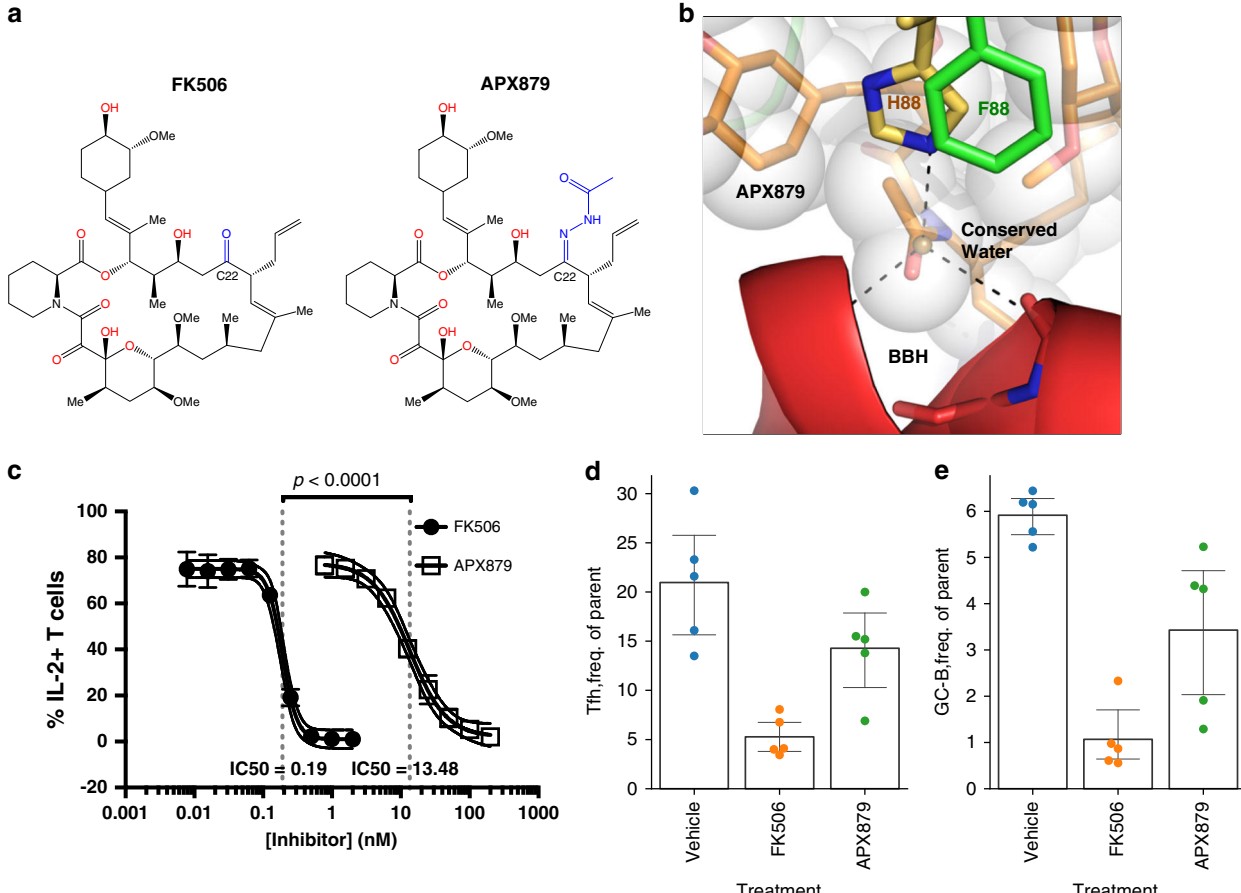

**Fig. 6** The FK506 analog APX879 has reduced immunosuppressive activity. **a** Chemical structures of FK506 (left) and APX879 (right) with modified C22 indicated in blue. **b** Model of APX879 bound to *A. fumigatus* CN–FKBP12 complex. APX879 was overlaid on FK506 in the *A. fumigatus* crystal structure and steric clashes with the acetohydrazine substitution were minimized manually. The oxygen atom donated by the substitution was placed such that it occupies a similar space to the conserved water bound to the BBH. **c** Naive primary murine CD4$^+$ T cells (from C57BL/6 mice) activated and differentiated in vitro and treated with FK506 have a reduced fraction of IL-2-producing cells than those treated with APX879. Dose response curves demonstrate an over 70-fold decrease in IL-2 production for FK506 compared to APX879. Note that the plot is showing a 95% confidence interval for the four-parameter curve fit, error bars for individual points represent the standard error value. The IC50 range is 0.17–0.21 for FK506 and 10.7–16.9 for APX879. **d**, **e** T helper cells (**d**) and germinal center (GC) B cells (**e**) collected from lymph nodes of mice treated for 8 days with vehicle, 5 mg/kg FK506, or 20 mg/kg APX879. Lymph nodes were collected 7 days following immunization with NP-OVA to stimulate T cell-dependent GC-B cell response. Error bars indicate boot strapped 95% confidence intervals. See also Supplementary Fig. 9

synthesized and screened (Supplementary Table 3 and Supplementary Fig. 8), and evaluated for their immunosuppressive activity in Jurkat cells[19,20]. One analog, APX879, with a C22-acetohydrazine substitution, showed reduced immunosuppressive effect with some antifungal activity, suggesting that the acetohydrazine moiety interacts less favorably with His88 of the mammalian complex when compared to Phe88 in the fungal complex (Fig. 6a, b). APX879 exhibited in vitro antifungal activity against the major pathogenic fungi *A. fumigatus, C. albicans, C. neoformans, M. circinelloides f. lusitanicus,* and *M. circinelloides f. circinelloides* (Supplementary Table 4). Additionally, strains that are FK506 resistant through mutations or deletion of FKBP12 or mutations in CnA or CnB were similarly resistant to APX879, validating that APX879 binds FKBP12 and functions via a similar mechanism as FK506 (Supplementary Table 5). Although APX879 was not as potent as FK506 in antifungal activity, consistent with FK506, APX879 also displayed synergistic antifungal activity with other antifungals examined in the different fungal species (Supplementary Table 6).

To assess the in vitro immunosuppressive effect of APX879 in comparison to FK506, naive primary murine CD4$^+$ T cells were

activated and differentiated in vitro, exposed to APX879 or FK506, and assessed for IL-2 production. Cells treated with APX879 exhibited a 71-fold reduction in immunosuppressive activity compared to FK506 (Fig. 6c). Additionally, the in vivo immunosuppressive activity of APX879 was compared to that of FK506 in mice immunized with NP-OVA. C57BL/6 mice were treated with vehicle, 20 mg/kg APX879, or 5 mg/kg FK506 24 h prior to immunization with NP-OVA and continued daily for 8 days. Animals treated with APX879 compared to FK506 mounted a significantly higher T helper cell-dependent, germinal center (GC) B cell response to the antigen demonstrating that APX879 is reduced in its in vivo immunosuppressive activity (Fig. 6d, e). The potential in vivo therapeutic efficacy of APX879 was assessed in various murine fungal infection models. In the cryptococcal infection model, both APX879 and fluconazole combination treatment regimens significantly extended median survival from 26 or 28 days to 33 and 34 days post infection ($P <$ 0.001) compared to either fluconazole or APX879 monotherapy (Fig. 7b), but did not significantly differ from each other ($P > 0.4$). Supporting the improved survival data, fluconazole treatment compared to vehicle treatment reduced fungal burden in three

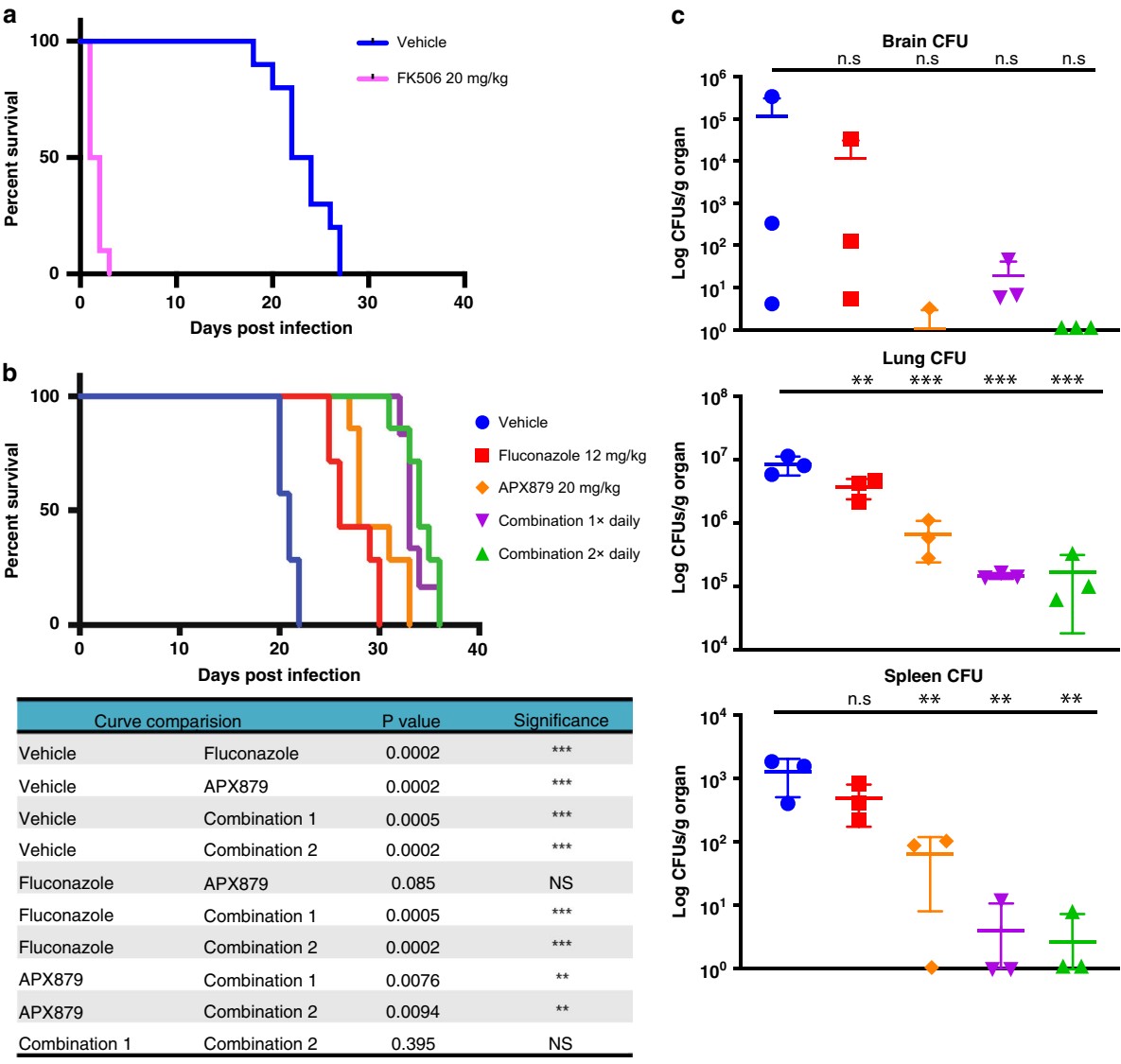

**Fig. 7** Efficacy of APX879 in cryptococcal model of murine invasive fungal infection. **a**, **b** Survival curve of mice infected intranasally with *C. neoformans* WT (H99) and treated daily with indicated antifungal drugs. Treatment was halted at day 14 and animals were monitored for survival. Survival curves analyzed using log-rank Mantel–Cox test. **c** Fungal burdens in brain, lung, and spleen of H99 infected and drug-treated animals. **P < 0.01, ***P < 0.001. Lower error bars that exceeded the *y*-axis limits were excluded. See also Supplementary Fig. 10

organs, with statistical significance in the lung (*P* < 0.01) (Fig. 7c). Although there was no statistical difference in the CFUs recovered from the brain in each treatment group due to high variability in the vehicle arm, the trend for the treatment groups matched those of the lungs and spleen. APX879 monotherapy, as well as the fluconazole combination treatment with APX879, significantly reduced disease burden in both the lungs and spleen (*P* < 0.001, 0.01 respectively) compared to vehicle-only treatment. Fluconazole combination treatment with APX879 is encouraging given the previous observation that the fluconazole combination with FK506 has an in vitro antifungal synergistic effect against *C. neoformans*[21]. Efficacy of APX879 was observed in the cryptococcal infection model, but not in the aspergillosis, mucormycosis, or systemic candidiasis models (Supplementary Fig. 10a–c). APX879 was efficacious in comparison to FK506 treatment (20 mg/kg) in the cryptococcosis model in which all the FK506-treated mice died by day 3, and 9 out of 10 mice died within 48 h following FK506 treatment (Fig. 7a). Lower doses of FK506 have been previously shown to be ineffective in a systemic cryptococcosis model[22]. The median survival of FK506-treated

animals was 1.5 days, while the median survival for vehicle-treated animals was 23 days. This acute toxicity of FK506 at 20 mg/kg in addition to therapeutic efficacy of the APX879 20 mg/kg dosing suggests that the modifications made to FK506 to produce APX879 allowed for the balance between the immunosuppressive and antifungal activities to open the therapeutic window compared to FK506. Although APX879 is reduced for both antifungal and immunosuppressive activity, our results suggest the possibility of designing more effective analogs that take advantage of these differences to be efficacious in an animal model.

## Discussion

Immunosuppression through CN inhibition is the major advance that made modern organ transplantation possible[3]. Selective targeting of CN is also a promising approach to design novel antifungals due to its pivotal cellular roles in diverse fungal pathogens[23]. However, engineering non-immunosuppressive and fungal-specific CN inhibitors requires a critical mechanistic

understanding of fungal CN inhibition. To identify differences between the human and fungal counterparts, we characterized the crystal structures of CN–FK506–FKBP12 complexes from four important human fungal pathogens (Fig. 1a, b). Binding and inhibition of fungal CN by the FKBP12–FK506 complex was analyzed through a combination of X-ray crystallography, NMR spectroscopy, site-directed mutagenesis, and MD simulation studies. The greater structural conservation between human and fungal CNs prompted an in-depth structural characterization of FKBP12s and their interaction with the CN complex.

Crystal structures revealed conserved and non-conserved motifs within FKBP12 across fungal and mammalian species with significant differences in the 40s and 80s loops (Fig. 1c, d). These differences do not appear to affect overall FK506 binding (Fig. 2), but impact the conformations adopted by the FKBP12–FK506 complex presented to CN (Fig. 3a, b). MD simulations provided further structural comparison of the *A. fumigatus* and human CN–FKBP12–FK506 ternary complexes under solution-like conditions and relaxation of crystal contacts. Counter-clockwise rotation of AfFKBP12 revealed a larger buried surface area between FKBP12–FK506 and CN, bringing AfFKBP12 residues Phe22, -Gln50, and -Arg55 into contact with CN, which are not observed in the X-ray structures. Strikingly, the opposite occurred in the human CN-complex, hFKBP12, which rotated clockwise compared to the *A. fumigatus* CN-complex. Despite this rotation in either complex, residues 20–55 and those in the 80s loop of FKBP12 still play pivotal roles in the interaction with CnA in the *A. fumigatus* and mammalian ternary complexes.

X-ray structures and MD simulations confirmed that the FK506 resistance of AfFKBP12-F22T, -Q50M, and -F88H is due to altered interactions with *A. fumigatus* CN but not with FK506 (Fig. 2). Unlike the F22T and Q50M mutants that showed partial FK506 resistance through minor disruption of FKBP12 and BBH interactions, the F88H mutant showed greater FK506 resistance due to alteration in the FK506-binding pocket, a possible increase in entropy of the 80s loop, and conformational flexibility of FK506 within the binding pocket. However, these potential conformational changes do not seem to affect the ability of F88H to bind to FK506 (Supplementary Fig. 2a, b). Although the His88 residue in mammalian FKBP12 is crucial for interaction with CN[14], substitution of *A. fumigatus* Phe88 with His (F88H) increased FK506 resistance and reduced CN binding in vivo (Fig. 2b, c). MD simulations suggest that the AfFKBP12–F88H mutation reduced the affinity for *A. fumigatus* CN from 198 nM to 3.29 μM, further supporting the difference of the Phe88 positioning in fungal FKBP12 X-ray structures in comparison to the His88 residue in mammalian FKBP12 (Supplementary Fig. 2c). Moreover, the H88F mutation in hFKBP12 in complex with *A. fumigatus* CN suggested near WT affinity of AfFKBP12 in complex with *A. fumigatus* CN, 334–198 nM, respectively. Although the hFKBP12–FK506 complex binds to *A. fumigatus* CN at the septum (Fig. 3d), it is unable to inhibit CN presumably due to orientation differences between the hFKBP12 40s or 80s loop contact residues in addition to those of *A. fumigatus* CnA-BBH/CnB. Increased FK506 sensitivity of the hFKBP12-K48Q, M50Q, and K48Q-M50Q mutants is likely due to the K48Q mutation resulting in direct contact with CnB and the M50Q mutation stabilizing the interaction with the BBH and restoring *A. fumigatus* WT interactions with CN (Fig. 4).

MD simulations also suggest that AfFKBP12 binds CN through a combination of hydrophobic and electrostatic interactions, while hFKBP12 binds almost exclusively electrostatically due to key electrostatic charge differences in essential residues, as well as charge reversals leading to an altered electrostatic surface for binding CN (Supplementary Fig. 11a, b). MD simulations on

hFKBP12 mutants (H88F, T86R, and H88F–T86R) suggested affinity similar to the *A. fumigatus* WT interaction, and these mutants were able to recover FK506 sensitivity (Fig. 4). The hFKBP12–H88F–FK506 complex bound to *A. fumigatus* CN through a combination of electrostatic and hydrophobic interactions, albeit in different conformational states. The H88F mutation in hFKBP12 appears to alter the FK506-binding pocket by increasing the polar characteristics, surface area, volume, and depth, further leading to an increase in entropy of the 80s loop and the conformational flexibility of FK506 within the binding pocket. The increased FK506 sensitivity of the hFKBP12–T86R and the T86R–H88F double mutant is possibly due to the increased positive electrostatic characteristic rendered by the T86R mutation. MD simulations on unbound and chimeric bound H88F mutation of hFKBP12 revealed that residues Arg41, Lys45, and Lys48 become more solvent exposed, generating a stronger positive electrostatic conformation to bind to CN. In the WT *A. fumigatus* CN complex, FKPB12 residues within the 40s loop do have contacts with CnB, while those in the 50s region have contacts with CnA (Fig. 5). However, MD simulations of hFKBP12 bound to WT *A. fumigatus* CN revealed that residues 53 and 54 of FKBP12 do not interact with the CN complex, while residues 41 and 45 do contact residues with CnB.

We provide general and specific mechanisms of action with respect to binding of FKBP12–FK506 to CN, highlighting key differences in the 40s and 80s loop in both *A. fumigatus* and mammalian ternary complexes. Stabilization of the FKBP12 80s loop is critical for packing of CN–FK506–FKBP12 complementary surfaces, which includes the key Phe88 residue specific to fungal FKBP12s. Structurally, FK506 contains two main parts: (i) a cyclohexane ring, a tricarbonyl group and a pipecolate moiety, that contact FKBP12, and (ii) an exposed effector region with C15, C18, and C21 moieties that are critical for interaction with CN[24] (Fig. 6a). Based on structural model, the His88 residue in hFKBP12 approaches FK506 ~1 Å closer than the Phe88 residue in AfFKBP12 and presents the possibility of adding an extension to FK506 at the O9 position on the C22 carbonyl for specific targeting of fungal CN (Fig. 6b). This difference highlighted in the X-ray and MD-simulated structures led to the design and synthesis of a less immunosuppressive FK506 analog, APX879 (Fig. 6c, d). Although we could demonstrate complex formation between *A. fumigatus* CN and FKBP12 in the presence of APX879 by recombinant expression, we have thus far been unable to crystallize the complex using approaches that previously led to the FK506 complex crystal structure.

Chemical modifications of FK506 have been previously studied[19,25,26] and the selective formation of both C22-oxime and hydrazone using unsubstituted hydroxylamine and hydrazine, respectively, has been reported[27,28]. We were able to synthesize 10 g of APX879 with highly efficient medical chemistry for robust characterization and use in animal infection models. Previously, most FK506 analogs with substitutions at the C15, C18, C21, C32 positions exhibited non-immunosuppressive activity because the FKBP–ligand complexes do not interact with or inhibit CN[24,29]. However an analog L-685,818, a C18-hydroxy and C21-ethyl derivative of FK506, did not inhibit mammalian CN[30] but inhibited *C. neoformans* CN in vitro[31], reflecting fungal versus mammalian CN–FK506–FKBP12-complex differences that can be exploited to design fungal-specific CN inhibitors. However, L-685,818 was not effective against *A. fumigatus* CN[32]. Recently other FK506 analogs synthesized were efficacious only in combination with FK506[19]. Another FK506 analog, 9-deoxo-31-*O*-demethyl–FK506, synthesized only in milligram quantities using engineered *Streptomyces* species was not efficacious as monotherapy but was synergistic with fluconazole to treat cryptococcal infection[22]. APX879 demonstrated broad-spectrum in vitro

antifungal activity, and different fungi with CN or FKBP12 mutations were resistant to APX879 (Supplementary Tables 4 and 5), validating a similar mechanism of action of APX879 as that of FK506 on CN. Our findings highlight the importance of structural approaches to the rational design and development of fungal-specific calcineurin inhibitors. However, we need to overcome the challenges of immunosuppressive activity of these inhibitors with simultaneous improvement in efficacy. We have shown that APX879 was synergistic with current antifungals and the possibility of using APX879 in combinatorial strategies to combat invasive fungal infections. Compared to FK506, APX879 exhibited reduced immunosuppressive activity both in vitro and in vivo, in vivo therapeutic efficacy in a murine cryptococcosis model, and a 17-fold reduced in vitro efficacy in *Cryptococcus* antifungal activity (Figs. 6 and 7). The benefits of CN inhibition for organ transplantation have been well-known for decades, and the CN pathway has been well established as critical for fungal pathogenesis and CN inhibition as an effective antifungal strategy in vitro[10]. APX879, reduced in both immunosuppressive and antifungal activity, is a proof-of-concept molecule that emphasizes that the balance between mammalian and fungal calcineurin inhibition can be disrupted to favor antifungal activity and be effective in vivo. These findings strengthen the paradigm for the development of non-immunosuppressive CN-specific inhibitors as antifungal agents for future therapeutics.

## Methods

**Experimental models.** *E. coli* DH5α competent cells were used for subcloning experiments. *A. fumigatus* WT strain *akuB*[KU80] was used for growth and mutagenesis experiments. *A. fumigatus* WT or recombinant strains were cultured on glucose minimal media (GMM) or RPMI liquid media at 37 °C for specified time periods. In certain experiments, GMM agar or RPMI liquid media were supplemented with FK506 (0.1–10 μg/ml). All growth experiments were repeated as technical triplicates, each also as biological triplicates. For in vitro antifungal susceptibility testing assays, the various fungal pathogens (*C. neoformans* WT (H99), *M. circinelloides f. lusitanicus*, *M. circinelloides f. circinelloides*, and *C. albicans*) were grown overnight in YPD, washed, and plated out to a lawn on YPD for *C. neoformans* and *M. circinelloides f. lusitanicus* strains or YPD + 10 mg/ml fluconazole for *C. albicans*.

For in vivo efficacy experiments with *C. neoformans*, female A/J mice (Jackson Labs 000646) were infected intranasally with 10^5 cells of *C. neoformans* H99. For in vivo experiments with *A. fumigatus*, male and female CD1 mice were immunosuppressed with cyclophosphamide (150 mg/kg) on days −2 and +3 of infection and triamcinolone (40 mg/kg) on days −1 and +6 of infection, with mice infected by intranasal inoculation (40 μl of 2.5 × 10^7 conidia/ml) *A. fumigatus* CEA10.

**Protein expression and purification.** For purification of the *A. fumigatus* and *C. immitis* CnA/CnB/FKBP12 protein complexes, triple expression constructs generated were expressed using the baculovirus expression system. *C. albicans* CnA–CnB fusion and the FKBP12 constructs were expressed separately in *E. coli* BL21 cells (DE3). For the *C. neoformans* CnA–CnB–FKBP12 protein complex, three single-ORF plasmids were generated and expressed using the baculovirus expression system. Briefly, protein expression was induced by isopropyl β-D-1-thiogalactopyranoside (IPTG) and Ni^2+ HiTrap chelating HP columns purified proteins. Purified proteins were concentrated, frozen, and stored at −80 °C until crystallography screening.

**A. fumigatus CN/FKBP12 complex.** A single triple expression plasmid DNA construct for baculovirus expression was cloned as follows, such that the final expression construct contained *A. fumigatus* CnA under the control of a polyhedrin promoter, *A. fumigatus* CnB under a P10 promoter, and *AfFKBP12* under a second polyhedrin promoter. For the CnA insert, residues 2–370 of UniProt Q4WUR1 were fused at the N-term in silico with residues MGSSHHHHHHSSGLVPRGS, and the gene insert was codon optimized for baculovirus expression by GeneComposer[33] (excluding restriction sites BglII, HindIII, EcoRI, XbaI, and StuI). The resulting CnA insert was flanked with 5′ BglII and 3′ HindIII, preceded by bases "TGATAA" to confer two stop codons. For the CnB insert, an initiating methionine was added to the N-term of residues 21–193 from UniProt Q4WDF2, and the insert was optimized for baculovirus expression by GeneComposer (excluding restriction sites BamHI, BglII, HindIII, EcoRI, XbaI, and StuI). The CnB gene was then flanked with 5′ BglII and 3′ EcoRI site preceded by bases "TGATAA" to confer two stop codons. For the FKBP12 insert, residues 2–112 of UniProt Q4WLV6 were fused at the N-term in silico with residues

MGLNDIFEAQKIEWHE, followed by codon optimization of the insert for baculovirus expression in GeneComposer (excluding restriction sites BamHI, BglII, HindIII, EcoRI, XbaI, and StuI). The FKBP12 insert was flanked with 5′ XbaI and 3′ StuI, preceded by bases "TGATAA" to confer two stop codons. The three genes for CnA, CnB, and FKBP12 were synthesized by GeneArt, then cloned in series into vector pBacgus4X-1 as follows: CnA was cloned first by digestion with BglII/HindIII followed by gel-purification, then ligated into BamHI/HindIII-digested vector and a resulting clone was confirmed positive by PCR screen. The CnB insert was then cloned into pBacgus4X-1/CnA via digestion of vector and insert with BglII/EcoRI followed by gel purification and ligation, and a resulting clone was confirmed positive by PCR screen. WT FKBP12 was also cloned into pBacgus4X-1/CnA–CnB via digestion of vector and inserted with XbaI/StuI followed by ligation. Following confirmation of the presence of FKBP12 by PCR screen, the ORFs for all three genes were confirmed by DNA sequencing.

To generate the *A. fumigatus* CnA, CnB, and FKBP12 complex, we first transfected pBacgus4X-1/CnA–CnB into Tni insect cells (Expression Systems) to generate isolated CnA–CnB heterodimer. Transfection was performed using BestBac 2.0, v-cath/chiA Deleted Linearized Baculovirus DNA (Expression Systems). Virus from each transfection was amplified through four rounds to produce viral stock for large-scale production. The large-scale preparations were grown in ESF921 medium (Expression Systems) of ~2.5 × 10^6 cells/ml that was infected with 5% final virus (37.5 ml) and harvested 72–96 h post-infection. For generation of the WT FKBP12 complex, we transfected pBacgus4x-1/CnA–CnB–FKBP12 as above and added FK506 to a working concentration of 5 μM 24 h post infection. Cells were centrifuged to collect cell paste that was frozen dropwise into liquid nitrogen and stored at −80 °C.

Insect cells expressing either CnA–CnB or CnA–CnB–FKBP12 were lysed via hypotonic lysis in 25 mM Tris pH 8.0, 0.02% CHAPS, 4.5 protease inhibitor tablets (ethylenediaminetetraacetic acid (EDTA) free) (Roche), the latter of which was supplemented with 5 μM FK506. The lysate was clarified via centrifugation at 42 K RPM (Beckman) for 45 min at 4 °C and filtered with a 0.2 μm bottle top filter (Nalgene). The supernatant was adjusted to 200 mM NaCl and was then applied to two Ni^2+ charged HiTrap Chelating HP (GE Healthcare) columns and the complex was eluted with a gradient of 25 mM Tris pH 8, 200 mM NaCl, 500 mM Imidazole, and 1 mM TCEP. Fractions containing the complex were pooled and the target protein was concentrated via centrifugal concentration (Vivaspin 20 Polyethersulfone (PES), 10 kDa MWCO, Sartorius). The concentrated sample containing the complex was purified via size exclusion chromatography using a Superdex 200 16/600 pg (GE Healthcare) column in 25 mM Tris pH 8.0, 250 mM NaCl, 1 mM TCEP. Fractions containing the complex were pooled, concentrated via centrifugal concentration as previously described to 10 mg/ml. Both complexes were flash frozen in liquid nitrogen for storage at −80 °C and set up immediately for crystallization trials.

**C. albicans CN/FKBP12 complex.** Plasmid DNA constructs for the expression of the CnA–CnB fusion proteins and FKBP12 were prepared as follows. For the *C. albicans* CnA–CnB fusion construct, residues 54–445 of UniProt Q5ABP4 (CnA) were fused in silico on the C-term to residues 1–173 of UniProt C4YS24 (CnB), flanked at the 5′ end by BamHI and at the 3′ end by bases TGAT, followed by a HindIII site (creating a double stop codon after the gene). Codons were optimized by GeneComposer for *E. coli* expression (excluding BamHI and HindIII sites from the coding region) and the resulting gene (CnA–CnB–2Xstop) was synthesized by GeneArt. The gene insert was cloned into a customized pET28a-based vector containing MGSSHHHHHHSSGLVPR, where the underlined "MG" corresponds to bases "ATGG" within the unique NcoI site in pET28a, followed by bases "GC". Open-reading frames (ORFs) for resulting clones with gene insert were then validated by DNA sequencing. The FKBP12 construct had residues 1–124 of UniProt P28870 flanked with 5′ NdeI and 3′ XhoI (preceded by bases "TAA" to confer a stop codon), and then cloned via NdeI/XhoI into pET15b.

The *C. albicans* calcineurin A–B fusion was transformed into *E. coli* BL21 cells (DE3). A starter culture containing 50 μg/ml (final concentration) of kanamycin (Teknova) was inoculated with a single colony and grown 16 h at 37 °C in terrific broth (TB) (Teknova). This was then transferred to 8 l of TB containing 50 μg/ml of kanamycin and was grown to OD_{600} = 0.6. Protein expression was induced by adding 1 mM IPTG (GoldBio) and the cultures were grown for 16 h at 20 °C. The cells were harvested by centrifugation (Beckman) at 5 K RPM for 15 min and the pellets were collected and stored at −80 °C.

Cells were lysed via sonication (Misonix) in 25 mM Tris pH 8.0, 200 mM NaCl, 50 mM L-arginine monohydrochloride, 0.02% 3-[(3-cholamidopropyl) dimethylammonio]-2-hydroxy-1-propanesulfonate (CHAPS), 1 mM Tris(2-carboxyethyl)phosphine (TCEP) (VWR), 100 mg Lysozyme, 250 U Benzonase (Novagen), and 1 protease inhibitor tablet (ethylenediaminetetraacetic acid (EDTA) free) (Roche). The lysate was clarified via centrifugation at 42 K RPM (Beckman) for 40 min at 4 °C and filtered with a 0.2 μm bottle top filter (Nalgene). The supernatant was applied to three Ni^2+ charged HiTrap Chelating HP (GE Healthcare) columns and the protein was eluted with a gradient of 25 mM Tris pH 8, 200 mM NaCl, 500 mM Imidazole, and 1 mM TCEP. The protein of interest was pooled, frozen in liquid nitrogen, and stored at −80 °C until complex formation.

*C. albicans* FKBP12 was transformed into *E. coli* BL21 cells (DE3). A starter culture containing 50 μg/ml (final concentration) of kanamycin and 100 μg/ml

(final concentration) of ampicillin was inoculated with a single colony and grown 16 h at 37 °C in terrific broth (TB) (Teknova). This was then transferred to 6 l of TB containing 50 μg/ml of kanamycin and 100 μg/ml of ampicillin and was grown to $OD_{600} = 0.8$. Protein expression was induced by adding 0.5 mM IPTG and the cultures were grown for 4 h at 37 °C. The cells were harvested by centrifugation (Beckman) at 5 K RPM for 15 min and the pellets were collected and stored at −80 °C.

Cells were lysed via sonication (Misonix) in 25 mM Tris pH 8.5, 200 mM NaCl, 50 mM L-arginine monohydrochloride, 0.02% 3-[(3-cholamidopropyl) dimethylammonio]-2-hydroxy-1-propanesulfonate (CHAPS), 1 mM Tris(2-carboxyethyl)phosphine (TCEP) (VWR), 0.25% glycerol (VWR), 100 mg lysozyme, 250 U benzonase (Novagen), and 1 protease inhibitor tablet (ethylenediaminetetraacetic acid (EDTA) free) (Roche). The lysate was clarified via centrifugation at 35 K RPM (Beckman) for 30 min at 4 °C and filtered with a 0.2 μm bottle top filter (Nalgene). The supernatant was applied to two $Ni^{2+}$ charged HiTrap Chelating HP (GE Healthcare) columns and the protein was eluted with a gradient of 25 mM Tris pH 8.5, 200 mM NaCl, 500 mM Imidazole, and 1 mM TCEP. The affinity tag (N-terminal His tag) was removed via cleavage with thrombin (BioPharm) while dialyzing against 25 mM Tris pH 8.5, 200 mM NaCl, 1 mM TCEP, 50 mM L-arginine monohydrochloride and 0.25% glycerol for 4 h at 4 °C. The affinity tag was removed by applying the digested pool over one HiTrap Benzamidine FF (high sub) (GE Healthcare) and two $Ni^{2+}$-charged HiTrap Chelating HP (GE Healthcare) columns in tandem and the target protein flowed through as anticipated. The target protein was concentrated via centrifugal concentration (Vivaspin 15 Polyethersulfone (PES), 5 kDa MWCO, Sartorius) pooled, frozen in liquid nitrogen, and stored at −80 °C until complex formation and crystallography screening.

**Complex formation**. Complexing reactions were performed by dialyzing the purified Calcineurin A–B fusion (at 90 μM) against 25 mM Tris, pH 7.5, 50 mM NaCl, 5 mM $CaCl_2$, and 0.5 mM TCEP. In parallel, FKBP12 was diluted to 50 μM by dilution with 25 mM Tris, pH 7.5, 50 mM NaCl, 5 mM $CaCl_2$, 0.5 mM TCEP, and 75 μM FK506 (1.5×) and incubated at 22 °C for 30 min. Samples of Calcineurin A–B and FKBP12/FK-506 were combined; the final concentration of calcineurin A–B was 28.5 μM and FKBP12/FK506 was 34.2 μM, or 1.2× FKBP12/FK506 over calcineurin A–B. The combined sample was incubated at 22 °C for 30 min before concentrating via centrifugal concentration (Vivaspin 20 Polyethersulfone (PES), 10 kDa MWCO, Sartorius) to 20 mg/ml. The concentrated sample containing the complex of CnA–CnB, FKBP12, and FK506 was purified via size-exclusion chromatography using a Superdex 200 10/300 GL (GE Healthcare) column in 10 mM Tris pH 7.5, 50 mM NaCl, and 1 mM $CaCl_2$. Three peaks eluted from the size-exclusion column, one of which was associated with the stoichiometric complex. The fractions containing the stoichiometric complex were pooled, concentrated via centrifugal concentration as previously described to 10 mg/ml.

***Coccidioides immitis* (strain RS) CN/FKBP12 complex**. One triple expression plasmid DNA construct for baculovirus expression was cloned and optimized similarly to the *A. fumigatus* CnA/CnB/FKBP12 triple expression construct, such that the final expression construct contained *C. immitis* CnA under the control of a polyhedrin promoter, *C. immitis* CnB under a P10 promoter, and *C. immitis* FKBP12 under a second polyhedrin promoter, except for the following differences. For the CnA insert, residues 27–395 of NCBI accession XP_001242771 were fused at the N-terminus with residues MGSSHHHHHHSSGENLYFQGS, and for the CnB insert, an initiating methionine was added to the N-terminus of residues 22–191 from UniProt A0A0D8JSK0, and for the FKBP12 insert residues 2–120 of UniProt A0A0J6YH06 were fused at the N-terminus in silico with residues MWSHPQFEKGS, followed by codon optimization of the insert for baculovirus expression in GeneComposer[33] (excluding restriction sites BamHI, BglII, HindIII, EcoRI, XbaI, and StuI). Genes were ordered from GeneArt.

Cloning into pBacgus4X-1 included two steps; first, the pBacgus4X-1 vector was digested with EcoRI/BglII/StuI/XbaI and the resulting 7805 bp vector backbone (EcoRI/XbaI-flanked) and 248 bp vector fragment (BglII/StuI-flanked, containing a p10 promoter) were gel-purified (other fragments were discarded). The genes for CnB and FKBP12 were digested with EcoRI/BglII and StuI/XbaI respectively, and digested gene fragments were gel-purified. The two gene fragments and the 248 bp vector fragment were then four-way ligated with the EcoRI/XbaI vector backbone, resulting in an intermediate construct. A positive clone for the intermediate construct was confirmed by PCR screening, then digested with BamHI/HindIII and gel-purified. The CnA gene was then digested with BglII/HindIII, gel-purified, and ligated into the BamHI/HindIIII-digested intermediate construct. One positive clone was then sequenced over all three open-reading frames.

The *C. immitis* CnA, CnB and FKBP12 construct was expressed as described for the production of *A. fumigatus* CnA–CnB–FKBP12. Purification of the complex through the first $Ni^{2+}$ purification step was also identical to that of *A. fumigatus*; however, following $Ni^{2+}$ purification, fractions containing the complex were pooled and the target protein was diluted three-fold to reduce the NaCl concentration to about 70 mM. The diluted pool was applied to tandem anion/ cation exchange chromatography (QFF and SP from GE Healthcare). Both columns were washed, the SP column was removed and the QFF column was eluted with a gradient of 25 mM Tris pH 8, 1 mM TCEP, 1 M NaCl, and 5 μM FK506. Fractions containing the complex were concentrated via centrifugal concentration (VivaSpin Turbo 15 Polyethersulfone (PES), 10 kDa MWCO, Sartorius). The concentrated sample containing the complex was purified via size-exclusion chromatography using a Superdex Hi Load 200 16/60 (GE Healthcare) column in 25 mM 2-(N-morpholino)ethanesulfonic acid (MES) pH 6.4, 150 mM NaCl, 0.5 mM TCEP, 5 mM $CaCl_2$, and 5 μM FK506. The fractions containing the complex were pooled, concentrated via centrifugal concentration as previously described to 9.26 mg/ml. The protein was both flash frozen in liquid nitrogen for storage at −80 °C and set up immediately in crystallization trials.

***C. neoformans* (strain H99) CN/FKBP12 complex**. Three single-ORF expression plasmid DNA constructs for baculovirus expression were cloned as follows, such that *C. neoformans* CnA, *C. neoformans* CnB, and a *C. neoformans/C. immitis* FKPBP12 chimera were expressed under the polyhedrin promoter from separate plasmids. For the *C. neoformans* CnA insert, residues 34–402 from Unipirot O42773 were fused at the N-term in silico with residues MGSSHHHHHH HSSGENLYFQGS, and the gene was codon-optimized for insect expression by ATUM (formerly DNA2.0), excluding restriction sites BglII, HindIII, EcoRI, XbaI, and StuI. The resulting sequence was flanked with 5′ and 3′ SapI sites (including two stop codons prior to the SapI site) and using SapI, was cloned into a SapI-modified cloning vector based on pBacgus-1 (Novagen); using this design the insert was placed between BamHI and HindIII sites within the multiple cloning site of pBacgus-1. *C. neoformans* CnB was synthesized and cloned in the same manner, using residues 1–175 from Uniprot P0CM55 (construct was not tagged). For the *C. neoformans/C. immitis* FKPBP12 chimera, residues 9–21 and 68–72 of Uniprot O94746 were replaced by residues KEGNGVDKPVKGD and SLGEK, respectively, from Uniprot J3K1B6, and residues SKRA were added to the C-terminus of the construct. The resulting gene starting with Uniprot O94746 residue 2 was then fused at the N-term with residues MWSHPQFEKGS and codon optimized by GeneWiz for insect expression (excluding restriction sites BglII, HindIII, EcoRI, XbaI, and StuI from the insert). The resulting nucleotide sequence was flanked with 5′ BamHI and 3′ TGAT-HindIII, and upon gene synthesis completion was cloned into BamHI/HindIII-digested pBacgus-1. As with previously described construct, all open-reading frames were confirmed in their entirety through flanking restriction sites by Sanger DNA sequencing. Expression and purification of the *C. neoformans* CnA–CnB–FKBP12 complex was performed as described for *A. fumigatus* CnA–CnB–FKBP12 WT complex in the presence of FK506.

**Structure determination by X-ray crystallography**. The *A. fumigatus, C. albicans, C. immitis*, and *C. neoformans* CnA, CnB, and FKBP12/FK506 complexes were placed into sparse matrix crystallization trials using sitting drop vapor diffusion at both 5 and 10 mg/ml, and, in the case of *C. albicans* and *A. fumigatus*, in the presence and absence of excess FKBP12/FK506. All crystallization trials were set up over a period of 5 days at 16 °C. Diffraction data for all complexes were reduced and scaled with XDS/XSCALE. Structures were solved by iterative molecular replacement using a previously deposited structure of CnA, CnB, FKBP12, and FK506 (PDB:1TCO) or related structures and the programs Phaser[34] and Molrep[35], in the CCP4[36,37], and Phenix[38] program suites. Structures were refined using iterative cycles of TLS and restrained refinement with Phenix Refine and model-building using COOT[39] and validated using Molprobity[40] and deposited in the Protein Data Bank[41,42]. Diffraction data and refinement statistics are listed in Table 1.

Crystals of the *A. fumigatus* complex with FKBP12 were grown in an optimization matrix screen (Rigaku Reagents) based on the sparse screen PACT condition E9 (0.1 M HEPES/NaOH, pH 7.4, 0.2 M potassium/sodium tartrate, 22.27% w/v PEG 3,350) and cryo-protected with 20% ethylene glycol. The dataset was collected on 9/6/2014 at the synchrotron Canadian Light Source beamline 08-ID on a RAYONIX MX-300 CCD X-ray detector. A single copy of the complex was placed per asymmetric unit (PDB ID: 6TZ7).

Crystals of *C. albicans* CnA–CnB and FKBP12/FK506 were grown in Proplex (Molecular Dimensions) condition A10 and then optimized to 0.1 M Tris–HCl, 15.91% PEG 2000 MME, 0.1 M potassium chloride, and cryo-protected with 20% ethylene glycol. The dataset was collected on 6/27/2013 at the synchrotron APS beamline 21-ID-G (LS-CAT) on a RAYONIX MX-300 CCD X-ray detector. Two copies of the complex were placed per asymmetric unit (PDB ID: 6TZ6).

Crystals of *C. immitis* CnA, CnB, and FKBP12/FK506 were grown in sparse matrix screen Proplex condition D2 (0.1 M MES pH 6.5, 10% PEG 5000 MME, and 12% Propanol-1) and cryo-protected with 20% ethylene glycol. The dataset was collected on 2/25/2015 at the synchrotron APS beamline 21-ID-G (LS-CAT) on a RAYONIX MX-300 CCD X-ray detector. A single copy of the complex was placed per asymmetric unit (PDB ID: 5B8I).

Crystals of *C. neoformans* CnA, CnB, and FKBP12/FK506 were grown in an optimization screen matrix (Molecular Dimensions reagents) based on Morpheus II condition A3 (0.1 M MOPSO/0.1 M Bis–Tris pH 6.5, 10% w/v PEG 8000, 20% w/v 1,5 pentanediol, 30 mM sodium sulfate, 30 mM potassium sulfate, and 30 mM lithium sulfate) and cryo-protected with 20% ethylene glycol. The dataset was collected on 11/9/2016 at the synchrotron APS beamline 21-ID-F (LS-CAT) on a Marmosaic 225 CCD X-ray detector. Two copies of the complex were placed per asymmetric unit (PDB ID: 6TZ8).

**FKBP12 and calcineurin A mutations in *A. fumigatus*.** Homologous integration constructs for the expression of AfFKBP12 and CnA were generated using the plasmid, pUCGH[15,16], and primers listed in Supplementary Data 1 and 2. Mutagenized linearized constructs were transformed into the *A. fumigatus* akuB[KU80] strain and transformants selected with hygromycin B (150 μg/ml). Transformants were verified for homologous integration by PCR, and accuracy of mutation by sequencing, and visualized by fluorescent microscopy. hFKBP12 expression construct was codon optimized for expression in *A. fumigatus* and synthesized by GenScript and cloned into pSFS2A vector. All strains generated are listed in Supplementary Data 3.

For the construction of various *AfFKBP12* mutations, an 800 bp *fkbp12* promoter was PCR amplified using *A. fumigatus* genomic DNA as a template and cloned at the KpnI-BamHI sites on pUCGH to generate the pUCGH-Fkbp12promo plasmids. A 1006 bp *fkbp12* terminator was PCR amplified using *A. fumigatus* genomic DNA as a template and cloned at the SbfI-HindIII sites on pUCGH-Fkbp12promo to generate a pUCGH-Fkbp12promo-Fkbp12term plasmid. Next, site-directed mutagenesis of *AfFKBP12* residues in the 40s and 80s loop (F22T; Q50M; R55E; F88H) was performed using pUCGH-FKBP12 cDNA plasmid as a template. Briefly, in the first PCR, two fragments were amplified using complementary primers (with respective mutation) overlapping the *fkbp12* region to be mutated and the respective primers at the N and C-terminus of the *fkbp12*. No stop codon was introduced at C-terminal end of *fkbp12* to facilitate expression of the *gfp* fusion. Next, fusion PCR was performed using equi-proportional mixture of the two PCR products as templates and the final *fkbp12* mutated PCR fragment of 336 bp was amplified with primers at the N and C-terminal of *fkbp12*. Mutated *fkbp12* fragments were digested with BamHI and cloned into the pUCGH-Fkbp12promo-Fkbp12term plasmid. Mutated *fkbp12* genes were sequenced to confirm the mutation and linearized with KpnI to facilitate homologous integration.

Site-directed mutagenesis of the different residues in calcineurin A was performed using the pUCGH-cnaA plasmid[16] as a template. Briefly, in the first PCR, two regions were amplified using complementary primers (with respective mutation) overlapping the *cnaA* region to be mutated and the respective primers at the N and C-terminus of the *cnaA*. No stop codon was introduced at C-terminal end of *cnaA* to facilitate expression of the *gfp* fusion. Next, fusion PCR was performed using an equi-proportional mixture of the two PCR products as templates and the final *cnaA*-mutated PCR product of 1674 bp was amplified with primers GCNA-F2 and CnaA-R-Bam at the N and C-terminal of *cnaA*. Mutated *cnaA* regions were digested with BamHI and cloned into the pUCGH-CnaApromo-CnaAterm plasmid harboring the 778 bp *cnaA* promoter and 386 bp *cnaA* terminator to facilitate homologous integration. Mutated *cnaA* genes were sequenced to confirm the mutation and linearized with XbaI and HindIII for homologous integration. Linearized constructs were transformed into *A. fumigatus* akuB[KU80] strain and transformants selected with hygromycin B (150 μg/ml). Transformants were verified for homologous integration by PCR and fluorescent microscopy. Recombinant strains were sequenced to confirm *cnaA* mutations. The PCRs and sequencing data for all the strains are available in the Source Data file.

**Protein extraction and Western analysis.** *A. fumigatus* recombinant strains expressing respective WT and mutated forms of FKBP12-GFP fusion proteins were cultured in GMM liquid medium at 200 rpm for 24 h at 37 °C. Crude extracts were prepared by homogenizing the mycelia using liquid nitrogen in buffer A (50 mM Tris–HCl, pH 7.5, 150 mM NaCl, 50 mM KCl, 0.01% Triton X-100, 1 mM PMSF, and 1:100 protease inhibitor cocktail). Total cell lysate was initially centrifuged at 1200 rpm to eliminate the cell debris, and the supernatant obtained was further centrifuged (5000 rpm, 10 min at 4 °C)[16]. Approximately 50 μg of protein electrophoresed on a 4–20% SDS–polyacrylamide gel was transferred onto a poly-vinylidene difluoride membrane (PVDF; Bio-Rad) and probed with rabbit polyclonal The[TM] anti-GFP primary antibody (1 μg/ml; GenScript) and peroxidase-labeled rabbit anti-IgG (1:5000; Rockland) secondary antibody. Detection was performed using SuperSignal West Pico chemiluminescent substrate (Thermo Scientific). Unprocessed and uncropped film images are available in the Source Data file.

**Microscopy.** Conidia ($10^4$) from the recombinant strains of *A. fumigatus* were inoculated into 5 ml GMM medium and poured over a sterile coverslip (22 × 60 mm; No.1) placed in a sterile dish (60 × 15 mm). Cultures grown for 18–20 h at 37 °C were observed by fluorescence microscopy using an Axioskop 2 plus microscope (Zeiss) equipped with AxioVision 4.6 imaging software. Uncropped fluorescence microscopy images are available in the Source Data file.

**Antifungal activity of FK506 and APX879.** In vitro antifungal activity of FK506 and its analog APX879 was assessed in RPMI 1640 (Sigma-Aldrich) for *C. neoformans* WT (H99), *A. fumigatus* WT (AF293), *A. fumigatus* WT (CEA10), *M. circinelloides f. lusitanicus*, *M. circinelloides f. circinelloides*, and *C. albicans* WT (SC5314). Minimum inhibitory/effective concentrations (MIC/MEC) for each drug were measured using modified Clinical and Laboratory Standards Institute (CLSI) M38-A2 and M27-A3 standard in vitro antifungal susceptibility protocols.

For testing the susceptibility of calcineurin/FKBP12 mutants in the various fungal pathogens to FK506 and APX879, the drugs (stocked at 5 mg/ml) were spotted onto sterile disks (Becton Dickinson 231039) and dried before being placed face down on freshly inoculated plate. 25 μg of FK506 and APX879 was applied to disks for *C. neoformans* and *M. circinelloides f. lusitanicus*. 100 μg of APX879 was applied to *C. albicans* due to decreased sensitivity. *M. circinelloides f. lusitanicus* sensitivity was also confirmed by MIC testing.

**Culture preparations for antifungal susceptibility testing.** Overnight cultures of each strain (*C. neoformans* WT (H99), *M. circinelloides f. lusitanicus*, *M. circinelloides f. circinelloides*, and *C. albicans*) were grown in YPD, washed in sterile water, and diluted to 0.0005 OD600/ml in RPMI 1640 or 50% fetal bovine serum (FBS) for *C. albicans*. Drugs were suspended in DMSO. Serial two-fold drug dilutions were made in their respective vehicles at 50× concentrated to result in a final concentration range of 16 μg/ml–7.8125 ng/ml. Drugs dilutions were added to a 96-well microtiter cell culture plate (Falcon 353072) containing the 0.0005 OD600/ml in RPMI 1640 or FBS. Plates were incubated at 35 °C for *Mucor* species and 37 °C for *C. neoformans* and *C. albicans* without shaking and monitored for growth at 24 and 48 h. For testing the susceptibility of calcineurin/FKBP12 mutants in the various fungal pathogens (*C. neoformans* WT (H99), *M. circinelloides f. lusitanicus*, *M. circinelloides f. circinelloides*, and *C. albicans*) to FK506 and APX879, strains were grown overnight in YPD, washed, and plated out to a lawn on YPD for *C.neoformans* and *M. circinelloides f. lusitanicus* strains or YPD + 10 mg/ml fluconazole for *C. albicans*.

**APX879 efficacy in a murine model of *C. neoformans* infection.** Vehicle stock for APX879 efficacy testing was prepared by mixing Kolliphor EL (Sigma C5135) and 200 proof ethanol in a 1:1 mix and filter-sterilizing it through a 0.20 μm filter (Corning 431222). To prepare vehicle doses, vehicle stock was diluted to 10% in sterile PBS (Sigma D8537), aliquoted to microcentrifuge tubes, and kept at −20 °C. Fluconazole doses were aliquoted from IV fluconazole 2 mg/ml in 0.9% sodium chloride and kept at room temperature in the dark. APX879 powder was dissolved in vehicle stock (100%) to make a 100 mg/ml APX879 stock. APX879 was diluted to the final concentration of 2 mg/ml (twice daily) and 4 mg/ml (single dose) in 10% Kolliphor EL/ethanol 90% PBS and kept at −20 °C.

*C. neoformans* inoculum was prepared by growing an overnight YPD culture, washing in sterile PBS, determining the concentration with a hemocytometer, and diluting to $2 \times 10^6$ cells/ml. Groups of 10 female A/J mice (Jackson Labs 000646) (15–20 g) were infected by inhalation with $10^5$ cells of *C. neoformans* H99. For infection by inhalation, animals were anesthetized using isoflurane and then 50 μl of inoculum was dripped onto the upright nostrils of the animal. Once the entire inoculum was inhaled animals were monitored for recovery before being returned to the cage.

APX879 was diluted to the final concentration of 2 mg/ml (twice daily) and 4 mg/ml (single dose) in 10% Kolliphor EL/ethanol 90% PBS and kept at −20 °C. Drug dosing began 1 day post infection (dpi) for all treatment groups and were administered daily for 14 days intraperitoneally. Following drug dosing on a daily basis, animal weight and health were assessed daily and doses were calculated for each individual based on weight. Study groups consisted of fluconazole 12 mg/kg, APX879 20 mg/kg daily (10 mg/kg twice daily), combination treatment of fluconazole 12 mg/kg and APX879 (either twice daily 10 mg/kg or once daily 20 mg/kg), and vehicle at the dosing volume appropriate for the combination group based on the animal's weight. On day 15 post infection, three animals from each group were randomly selected for euthanasia and organs harvested for fungal burden analysis. Remaining animals were monitored in a blinded manner for daily survival until all animals reached the humane endpoint. Survival curves were generated using GraphPad Prism 7 and analyzed with a log-rank (Mantel–Cox) test.

**Cryptococcal fungal burden analysis.** To assess fungal burden, animals were euthanized by $CO_2$ inhalation on day 15 post infection. Organs (brain, lung, and spleen) from each animal were harvested and homogenized in sterile PBS using steel beads (Qiagen 69989) and a bead beater. Each organ homogenate was 10-fold serially diluted to $10^{-4}$ in PBS. 100 μl of each dilution was plated to antibiotic plates (YPD 50 μg/ml Amp 30 μg/ml chloramphenicol), incubated at 30 °C, and assessed for colony forming units. CFU per organ weight was calculated and plotted using GraphPad Prism 7. Statistical significance was calculated using an ordinary one-way ANOVA with Dunnett's multiple comparisons test.

**Murine model of invasive aspergillosis.** Six-week-old CD1 male mice (mean weight 22.5 g) were immunosuppressed with both cyclophosphamide (150 mg/kg on day −2 and 100 mg/kg on day +3) and triamcinolone acetonide (40 mg/kg on days −1 and +6). Four groups of 20 mice each were immunosuppressed and intranasally infected with the *A. fumigatus* CEA10 WT strain (40 μl of $3 \times 10^7$ spores/ml) and received either no treatment, vehicle only, APX879 (20 mg/kg once daily IP), or FK506 (1 mg/kg once daily IP). FK506 was not delivered at APX879 equipotent dosing, due to earlier toxicity studies that showed rapid death of all mice when higher doses (>1 mg/kg daily) utilized. Survival was plotted on a Kaplan–Meier curve and log rank was used for pair-wise comparison of survival

with statistical significance defined as a two-tailed $P < 0.05$. Histopathological examination of the lungs was performed in two mice in each group that were euthanized on day +4 of infection. Lungs were embedded in 10% neutral buffered formalin and subsequently sectioned and stained with Gomori methenamine silver and hematoxylin–eosin for assessment of histological signs of infection. The animal model and experiments were conducted in accordance with the Animal Care and Use Program of the Duke University Medical Center.

**Murine model of invasive mucormycosis.** *M. circinelloides f. circinelloides* spores were prepared by inoculating YPD solid medium and collecting fresh spores after 4 days. Groups of five male BALB/c mice (20 g) were infected with $1.25 \times 10^6$ spores in sterile PBS. Mice were anesthetized with isoflurane and infected intravenously through retroorbital injection. Drug dosing was begun concurrently with infection and continued for 7 days. APX879 analog injections were diluted and administered intraperitoneally, as described for the *Cryptococcus* model. Liposomal amphotericin B (AmBisome) was reconstituted as per manufacturer instructions and administered intravenously through the retroorbital route. Animals were monitored in a blinded manner for weight and daily survival.

**Murine model of invasive candidiasis.** *C. albicans* wild type (SC5314) inoculum was prepared by growing an overnight YPD culture, washing in sterile PBS, determining the cell concentration with a hemocytometer, and diluting to $7 \times 10^6$ cells/ml. Groups of six male CD1 mice (20–25 g) were infected intravenously by retro-orbital instillation of $7 \times 10^5$ cells in 100 μl. Animals were anesthetized by isoflurane prior to infection. Animals were treated once daily for 6 days immediately following infection with either 20 mg/kg APX879, 0.25 mg/kg fluconazole, or a combination therapy of the same doses. Kidneys were harvested for fungal burden analysis as described in the *Cryptococcus* animal model.

**Cell culture.** Pooled spleen and lymph nodes from C57BL/6 mice were homogenized through a 40 μm filter and subjected to magnetic enrichment of naive CD4 + T cells using the eBioscience MagniSort mouse naive T cell kit (eBioscience/Invitrogen/ThermoFisher). Cells were grown in Iscove's modified Dulbecco's medium (IMDM) supplemented with glutamine, penicillin, streptomycin, gentamicin, 2-mercaptoethanol, and 10% FBS. Purified naive T cells were cultured on anti-hamster IgG-coated plates in the presence of hamster anti-CD3 epsilon and anti-CD28 antibodies (eBioscience), neutralizing anti-IL-4 antibody (eBioscience), recombinant IL-12 (10 ng/mL), and recombinant IL-2 (50 U/mL) for 72 h. Cells were cultured in the presence of serially diluted FK506 or APX879 suspended in DMSO during these 72 h. During the last 4 h of culture, phorbol 12-myristate 13-acetate (PMA; Sigma), ionomycin (Sigma), and GolgiStop (BD Biosciences) were added to the culture to facilitate detection of intracellular cytokines.

**Staining and flow cytometry.** Live cells were stained with anti-CD4 antibody (eBioscience) and Fixable Viability Dye eFluor 780 (eBioscience) on ice, followed by fixation and permeabilization using the Foxp3 Transcription Factor Staining Kit (eBioscience). Intracellular staining was performed at room temperature using anti-IL-2 and anti-IFN-γ (eBioscience), followed by analysis with a BD LSRFortessa X-20 flow cytometer. FACS data for Fig. 6c, d are provided in Source Data file.

**In vivo immunosuppression of FK506 and APX879.** Groups of five female healthy C57BL/6 mice were treated daily for 8 days with vehicle, 5 mg/kg FK506, or 20 mg/kg APX879 via IP injection. 24 h following initial treatment, animals were immunized with the antigen NP-OVA (100 μg) in the adjuvant (Alhydrogel) via subcutaneous instillation. Drug treatment continued daily for 7 days following immunization to allow for T cell-dependent Germinal Center (GC) B cell response. Animal lymph nodes were harvested 7 days following immunization and sorted to isolate populations of T helper cells and GC B cells. Abundance of T helper cells and GC B cells from each animal was determined using FlowJo software. FACS data for Fig. 6d is provided in Supplementary Fig. 9.

**NMR analyses.** Recombinant FKBP12 proteins were expressed as N-terminal 6X-His tagged proteins in *E. coli* BL21(DE3) cells grown at 25 °C in minimal M9 media supplemented with 1 g/l $^{15}NH_4Cl$ as the sole nitrogen source for $^{15}N$ labeled protein or 1 g/l $^{15}NH_4Cl$ and 2 g/l $^{13}C$-labeled D-glucose for doubly labeled protein. Cells were grown to an $A_{600}$ of 0.6 and induced with 1.0 mM IPTG for 4 h. Cells were harvested by centrifugation and the pellet was flash frozen. The FKBP12 proteins were purified using a Ni NTA column followed by overnight cleavage of the 6X-His tag by thrombin. The cleaved proteins were dialyzed to remove the imidazole and then run over a Ni NTA column again to remove the 6X-His tag. Subsequent size-exclusion chromatography using a sephacryl S100HR XK26/60 FPLC column yielded pure protein. All NMR experiments were performed at 25 °C on FKBP12 proteins in 20 mM phosphate, 100 mM NaCl, 0.02% NaN₃ at pH 6.0. hFKBP12–H88F and *AfFKBP12*–F88H mutant constructs were obtained from GenScript in the pET-15b vector.

NMR backbone and side-chain resonance assignments for the FKBP12 proteins from the pathogenic fungi *A. fumigatus* and *C. albicans* were determined in both the apo and FK506 bound states using reduced sampling NMR experiments

(HNCO, HN(CA)CO, HNCA, HN(CO)CA, HN(CA)CB, HN(COCA)CB, HA(CA) NH, HA(CACO)NH, 4D HCCCONH, and HCCH_TOCSY) and $^{15}N/^{13}C$ uniformly labeled proteins utilizing a Bruker Avance 600 MHz spectrometer. Chemical shifts were referenced to an external 2,2-dimethyl-2-silapentane-5-sulfonate (DSS) sample. Data were processed and analyzed using NMRPipe and NMRViewJ. The programs PINE and AutoAssign were used to determine chemical shift assignments. For comparison purposes, the resonance assignments of the human form of the protein with and without bound FK506 have also been determined in our laboratory using the same buffer and methods.

NMR-based FK506 titration studies were performed using the FKBP12 proteins from *A. fumigatus* and *H. sapiens*. FK506 titration experiments were executed using FKBP12 proteins at a concentration of 100 μM in all cases. The immunosuppressive drug, FK506, was dissolved in DMSO at a concentration of 6 mM. $^1H–^{15}N$ HSQC data were collected for titration points of 0, 0.1, 0.25, 0.5, 0.75, 1.0, and 2.0:1 FK506:FKBP12. This gives a final concentration of 456 mM DMSO in solution after the final titration point. Trial titrations were performed without FK506. Even at the highest concentration of DMSO, no chemical shifts were observed for the FKBP12 HSQC resonances indicating the suitability of this solvent. The $^1H$, $^{15}N$, and $^{13}C$ resonance assignments for the apo and FK506 bound *A. fumigatus* and Human FKBP12 proteins are available at BioMagResBank (http://www.bmbr.wisc.edu) under the accession codes 27732, 27733, 27738, and 27739. Raw NMR data is provided in Source Data file.

**Molecular modeling of CN–FK506–FKBP12 complexes.** FKBP12 and CN interface residues for mammalian (1TCO.pdb) and *A. fumigatus* complexes were determined in qtPISA[43]. MODELLER v9.18 was used to construct a model of the human ternary complex of CN–FK506–FKBP12 using the bovine ternary complex of CN–FK506–FKBP12 as the template (PDB: 1TCO)[7,44]. During the model building process, we employed an optimization method involving conjugate gradients and MD to minimize violations of the spatial restraints. 500 models were generated between an alignment of the human sequence and the bovine PDB and scored by the internal MODELLER scoring method discrete optimized protein energy (DOPE)[45]. The structure with the lowest DOPE score was subsequently run through PROCHECK and WHATCHECK (checks the stereochemical quality of a protein structure) for quality[46,47]. For the FKBP12 mutations, PDBs were constructed by PyMOL from either the X-ray characterized *A. fumigatus* or MODELLER-derived human PDBs (The PyMOL Molecular Graphics System, version 1.7.4; Schrödinger, LLC.).

For all molecular docked and simulated complexes default High Ambiguity Driven DOCKing (HADDOCK) parameters were used throughout the docking procedures[48] with the following exceptions: (i) random exclusion of ambiguous restraints was set to false, (ii) hydrogen bond restraints were used, (iii) dihedral restraint energy constants were used, (iv) sampling of 180° rotated solutions during iteration 0 (rigid docking) was set to false, and (v) calculation of explicit desolvation energy was set to true. The docking procedure consisted of a heterodimer FKBP12:FK506 complex docking to a heterodimer CnA/CnB complex. The human heterodimer complexes were generated as stated above, while the *A. fumigatus* heterodimer complexes were generated from the X-ray structure described above. Active residues (residues directly involved in the interaction) were defined as those residues directly involved in the interaction between FKBP12: FK506 and CnA/CnB in the X-ray characterized *A. fumigatus* ternary complex. For the human ternary complex docking, the active residues were determined through alignment with the *A. fumigatus* sequence. Residues immediately surrounding the defined active residues with 50% solvent accessibility were defined as passive residues (those residues indirectly involved in the interaction). One thousand structures were generated within the first rigid docking iteration, the 20% with the lowest HADDOCK scores were then further refined in a semiflexible in vacuuo environment, and all structures from this semiflexible iteration were further water refined in the final iteration. Each docking attempt was performed 10 times, and the solution with the lowest HADDOCK score was retained. The RMSDs of the complexes were calculated with the McLachlan algorithm[49] as implemented in ProFit (http://www.bioinf.org.uk/software/profit/). A cluster analysis of the final docking solutions was performed with a minimum cluster size of five. The cutoff for clustering was manually determined for each docking run. The RMSD matrix was calculated over the backbone atoms (C, N, HN, and Cα atoms) of the interfacing residues. All images were produced with PyMOL.

All MD simulations were performed with the GROMACS 5.0.1 software package utilizing six CPU cores and one NVIDIA Tesla K80 GPU to provide a solution representation of the ternary complex of CN–FK506–FKBP12[50]. The single starting conformations used for all of the MD simulations were the lowest energy structure from the highest populated cluster from the HADDOCK docking as defined above. MD simulations were performed with the AMBER 99sb-ildn force field and the flexible simple point-charge water model[51]. FK506 topology and parameter files were generated through the use of AMBER Antechamber[52]. The initial structures were immersed in a periodic water box with a dodecahedron shape that extended 1 nm beyond the protein in any dimension and neutralized with counterions. Energy minimization was accomplished through use of the steepest descent algorithm with a final maximum force below 100 kJ/mol/min (0.01 nm step size, cutoff of 1.2 nm for neighbor list, Coulomb interactions, and Van der Waals interactions). After energy minimization, the system was subjected

to equilibration at 300 K and normal pressure for 1 ns. All bonds were constrained with the LINCS algorithm and virtual sites were used to allow a 4 fs time step (cutoff of 1.4 nm neighbor list, Coulomb interactions, and Van der Waals interactions). After temperature stabilization, pressure stabilization was obtained by utilizing the v-rescale thermostat to hold the temperature at 300 K and the Berendsen barostat was used to bring the system to 1 bar pressure. Production MD calculations (500 ns) were performed under the same conditions, except that the position restraints were removed and the simulation was run for 100 ns (cutoff of 1.1, 0.9, and 0.9 nm for neighbor list, Coulomb interactions, and Van der Waals interactions). GROMACS built-in and homemade scripts were used to analyze the MD simulation results. In general, the last 100 ns of each simulation was used for analysis, which included clustering of similar conformational states into an average structure defined as the middle structure of the RMSD matrix. All images were produced with PyMOL. All scoring of protein–protein complexes were accomplished through FireDock[53]. Finally, the binding pocket analysis was accomplished through DoGSiteScorer[54].

**Synthesis of APX879 (FK506–C22–acetohydrazide).** To a solution of FK506 (100 mg, 0.12 mmol) in ethanol (2 ml), acetohydrazide (55.3 mg, 0.75 mmol) was added and the solution was stirred for 12 h at 90 °C. The mixture was concentrated under reduced pressure and partitioned between ethyl acetate (50 ml) and water (50 ml). The layers were separated and the aqueous phase was extracted twice with ethyl acetate (30 ml). The combined organic layers were washed in brine (30 ml), dried over sodium sulfate, filtered and concentrated under reduced pressure. The residue was purified by flash chromatography using ethyl acetate and hexane as eluent. Fractions containing the product were pooled and all solvents were removed under reduced pressure to give FK506–C22–acetohydrazide as a pale yellow residue (68 mg, 0.08 mmol, 64% yield). 400 MHz $^1$H NMR (CDCl$_3$); The signals observed in the spectrum of APX879 were consistent with the signals observed for spectrum of FK506 with the indicative appearance of a signal for the acetohydrazide methyl group CH$_3$ at δ 2.19; 125 MHz $^{13}$C NMR (CDCl$_3$); The signals observed in the $^{13}$C-NMR spectrum of 879 were consistent with the signals observed for spectrum of FK506 with the indicative absence of the C22-carbonyl signal C = O at 212.8 ppm and the appearance of signals for the acetohydrazide carbonyl C = O at δ 175.2 and the C22-hydrazide C = N at δ 157.1. High-resolution MS: 858.5130 [M–H]$^-$; calculated C$_{46}$H$_{72}$N$_3$O$_{12}$ [M–H]$^-$ = 858.5121. The $^1$H and $^{13}$C-NMR data and high-resolution MS for APX879 are provided in the Source Data file.

**Ethics statement.** Animal studies at Duke University Medical Center were in full compliance with all of the guidelines of the Duke University Medical Center Institutional Animal Care and Use Committee (IACUS) and in full compliance with the United States Animal Welfare Act (Public Law 98-198). Duke University Medical Center IACUC approved all of the vertebrate studies under the protocol numbers A-038-11-02, A171-16-08, and A019-18-01. The studies were conducted in the Division of Laboratory Animal Resources (DLAR) facilities that are accredited by the Association for Assessment and Accreditation of Laboratory Animal Care (AAALAC).

**Reporting summary.** Further information on research design is available in the Nature Research Reporting Summary linked to this article.

## Data availability

Crystal structures are deposited in the Protein Data Bank under PDB codes 6TZ6 [https://www.rcsb.org/structure/6TZ6], 6TZ7 [https://www.rcsb.org/structure/6TZ7], 6TZ8 [https://www.rcsb.org/structure/6TZ8], and 5B8I [https://www.rcsb.org/structure/5B8I]. The $^1$H, $^{15}$N, and $^{13}$C resonance assignments for the apo and FK506 bound *A. fumigatus* and human FKBP12 proteins have been deposited in BioMagResBank (http://www.bmrb.wisc.edu/) with accession codes 27732 [https://doi.org/10.13018/BMR27732], 27733 [https://doi.org/10.13018/BMR27733], 27738 [https://doi.org/10.13018/BMR27738], and 27739 [https://doi.org/10.13018/BMR27739]. The source data underlying Figs. 2a–c, 3c, d, 4a, 6a, c–e, Supplementary Figs. 1g, 2a, b, 5a, b, 6a, b and 7b, and Supplementary Tables 4–6 are provided as a Source Data file. A key resource table is also provided in the Source Data file. Other data supporting the findings of this study are available from the corresponding authors upon request.

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

## Acknowledgements

This work was supported by grants from the NIH/NIAID (R01 AI112595-04; P01 AI104533-04; R43AI098300; R43AI106235) and an Astellas Pharma Global Development Research grant. This work was in part supported by the General International Collaborative R&D program, funded by the Ministry of Trade, Industry and Energy (MOTIE) of the Republic of Korea (N0001720) (to J.H. and Yong-Sun Bahn). The authors thank Duke's Research Computing staff for the use of Duke Computing Cluster and its support and others at Beryllium Discovery for contributions to the research and review of the manuscript. Use of the Duke NMR Spectroscopy Center instrumentation is gratefully acknowledged. The authors thank Dr. Jiyong Hong and Hyunji Lee (Duke University) for providing the high-resolution mass spectra for APX879. Crystallization work was funded by the NIH/NIAID (contract nos. HHSN272200700057, HHSN272201200025C, and HHSN272201700059C to Peter J. Myler). This research used resources of the Advanced Photon Source, a US Department of Energy (DOE) Office of Science User Facility operated for the DOE Office of Science by Argonne National Laboratory under Contract no. DE-AC02-06CH11357. Use of the LS-CAT Sector 21 was supported by the Michigan Economic Development Corporation and the Michigan Technology Tri-Corridor (Grant 085P1000817). Research was performed on beamline 08ID-1 at the Canadian Light Source, supported by the Canada Foundation for Innovation, Natural Sciences and Engineering Research Council of Canada, the University of Saskatchewan, the Government of Saskatchewan, Western Economic Diversification Canada, the National Research Council Canada, and the Canadian Institutes of Health Research. Compound APX879 was supplied by Amplyx Pharmaceuticals, Inc. S.M.C.G. was the recipient of an NSERC postdoctoral fellowship.

## Author contributions

P.R.J. supervised, designed, and performed genetic, biochemical, and microscopy experiments; analyzed and interpreted data and wrote the manuscript. D.F. designed, performed protein expression, X-ray crystallography, analyzed and interpreted structural data, and contributed to writing. B.G.B. designed and performed MD simulations, structural modeling, analyzed and interpreted structural data, and contributed to writing. M.J.H. designed and performed cryptococcal murine infections, antifungal susceptibility and immunology experiments, and contributed to writing. S.M.G. interpreted structural data and performed structural modeling. R.A.V. designed, performed, and interpreted NMR experiments. Z.C., J.J.L., A.F.A., performed *Cryptococcus*, *Mucor*, and *Candida* infections, and antifungal susceptibility assays. D.C.C. and B.C.B. performed antifungal susceptibility, and *Aspergillus* infection experiments. J.D.W. and M.C. designed and assisted in immunology experiments. M.T., X.L. and M.M. designed and performed the synthesis of APX879. S.C.L. contributed to structural modeling. Y.-L.C. contributed to antifungal susceptibility assays with FK506 analogs. L.D.S. supervised NMR structural studies. M.A.S. contributed to structural data interpretation. J.H. and W.J.S. designed and supervised the overall study. All authors were involved in editing the manuscript.

## Additional information

**Competing interests:** The authors declare no competing interests.

