## [Peer Review File · Nature Communications]

Reviewers' comments:

Reviewer #1 (Remarks to the Author):

The manuscript details experimental (X-ray crystallography, NMR spectroscopy, site directed mutagenesis, in vitro activity, in vivo therapeutic efficacy in an infection model) and theoretical (model building, docking and molecular dynamics simulations) investigations towards the development of a less immunosuppressive FK506 analog, APX879, as novel effective antifungal therapeutics.

This work is an extension on previously reported work by the authors [most recently 'Structures of pathogenic fungal FKBP12s reveal possible self-catalysis function', (2016) MBio 7 e00492-e00416], however here the authors report the first crystal structures of the CN ternary complexes with FKBP12-FK506 from four highly clinical relevant human fungal pathogens.

This is an impressive structure/activity/functional study where some of the structural elements of the ternary complexes were recognized as essential for binding and inhibiting specificity of fungal versus mammals CN interactions, namely the key residue Phe88 of FKBP12 which is not conserved in mammalian FKBP12 (His88).

The manuscript is clearly written and read generally well, the experimental and theoretical methodology is appropriate and the results appear to be valid and are well presented, founded on a thorough structural and in vitro / in vivo biological data analysis.

However, I have two major concerns:

i) The development of APX879, a FK506 analog, based on the structural differences of the human and fungal ternary CN complexes, is an interesting and encouraging finding in the search of more potent and selective anti-fungal.

However, APX879 exhibits, on activated naïve CD4⁺T cells, just a 71-fold reduction in immunosuppressive activity compared to FK506 and it shows a limited therapeutic efficacy in the tested murine fungal infection models.

ii) The general significance and interest of this manuscript.

IMHO, it is too medicinal chemistry focused, for the diverse readership of Nature Communications.

Overall, I strongly recommend the authors to submit their manuscript to a more specialized chemical biology or medicinal chemistry focused journal, i.e. Cell Chemical Biology, ACS Chemical Biology, and ACS Journal of Medicinal Chemistry.

Here are some minor revisions needed:

The Abstract should directly mention the results for APX879

For the sake of clarity, the authors are kindly request to adopt throughout the manuscript a "unifying" numbering scheme for the protein residues. It should be consistent with those in the deposited related PDB files.

For instance, in the PDB entry 5HWB (*A. fumigatus* FKBP12) the numbering of selected residues is: Asp38, Arg43, Phe88 and Ile92, whereas in the PDB entries 1TCO (*B. taurus* CN-FKBP12-FK506 ternary complex), 2PPN (human FKBP12) and 1FKJ (human FKBP12-FK506 complex) the numbering is Asp37, Arg42, His87 and Ile90.

In the PDB entry 5b8i (*C. immitis* CN-FKBP12-FK506 ternary complex) the numbering instead is Asp56, Arg61, Phe105 and Leu108.

Page 6, line 145

"are close to FKBP12" should read "are close to FK506"

Page 31, Table 1.

In Table 1, from a methodological point of view, the Pearson correlation coefficients CC* and the

CC1/2 values instead of the R-merge residual should be provided. Reporting the indicator R-merge is quite a controversial issue. See: Karplus P.A, Diederichs K. "Assessing and maximizing data quality in macromolecular crystallography. *Curr Opin Struct Biol.* (2015) 34, 60-68; Diederichs K, Karplus PA. "Better models by discarding data?" *Acta Crystallogr D Biol Crystallogr.* (2013) 69, 1215-1222; Diederichs K., Karplus P.A. "Linking crystallographic model and data quality" *Science* (2012) 336, 1030-1033; Karplus P.A., Diederichs K. "Improved R-factors for diffraction data analysis in macromolecular crystallography" *Nat Struct Biol* (1997) 4, 269-275 and Erratum in *Nat Struct Biol* (1997) 4, 592.

The authors should distinctly specify in Table 1, the N° of atoms and ADP (Å²) values for CN-A, CNB, FKBP12 and FK506 respectively. Please specify the chemical nature of the "Ligand".

In Table 1 for each of the structures, should also be included the values representative of the Ramachandran plots (Favored / Outliers (%)) as well as those of the MolProbity validation analysis, i.e. Rotamer outliers (%) and the MolProbity scores.

Page 34, line 833

"of approximately 2.5e6 cells/ml " should read "of approximately 2.5·10⁶ cell/mL"

Page 41, line 1003

The authors indicate PDB entries 6BLS and 6BLU.

In Table 1, the crystallography and structure refinement statistics for the 6BLS PDB entry are absent.

Moreover the authors should clearly state that the 6BLS PDB entry is related to the crystal structure of the ternary complex of *A. fumigatus* Calcineurin A, Calcineurin B, the P90G FKBP12 mutant and FK506.

Page 42, line 1004

"were grown in Proplex condition" should read "were grown in Proplex (Molecular Dimensions) condition"

Page 42, line 1015

"screen matrix screen (Rigaku reagents) based on Morpheus II" should read "screen matrix (Molecular Dimensions reagents) based on Morpheus II"

Page 52, line 1249

The authors are kindly request to clarify the meaning of "during rigid body EM was set to false"

Figure 2, Legend

In the last sentence, the authors should clearly state the PDB entry 5HWB is related to the WT *A. fumigatus* FKBP12 crystal structure while the PDB entry 5HWC is related to the crystal structure of the *A. fumigatus* FKBP12 mutant P90G in complex with FK506.

Figure 6, Panel C

The IC50 values of FK506 and APX879 are given without any error estimates

Figure S1, Legend

Panel (F), the PDB entry PDB 1TCO is not human FKBP12 but rather *B. taurus* CN-FKBP12-FK506 ternary complex.

Figure S3

The entities and the related units (X- and Y-axes) have been omitted in all the plots (Panels A-E)

Figure S8

The authors should indicate in the legend that the molecular surface is colored according to the

calculated electrostatic potential contoured from $-XX$ kT/e (red) to $+YY$ kT/e (blue). The same apply for the hydrophobic surface plot.

Reviewer #2 (Remarks to the Author):

The paper by Juvvadi et al. reports the crystal structure of calcineurin catalytic and regulatory subunit complexed with FK506 and FKBP12 from the most common human fungal pathogens. Calcineurin is required for fungal growth and inhibitors of this complex, such as FK506 or Cyclosporin A, are fungicidal. However, these inhibitors are potent immunosuppressive in mammalian cells, limiting their use in the clinic against fungal infections. Using a structural biology approach, the authors found key residue(s) in the fungal enzyme complex not conserved in the human counterpart. This difference was used to design and synthesize derivatives of FK506, such as APX879, with higher affinity to the fungal complex. As a result, APX879 is much less immunosuppressive than FK506. Importantly, APX879 maintains the broad spectrum of antifungal activity of the parent compound and it is also efficacious in a murine model of invasive fungal infection. I found the studies of this paper convincing, well-illustrated and thoroughly discussed. I also found the experimental design well thought out with appropriate controls in place. I also think these studies are highly significant, and they will have a dramatic impact in the research and development of new antifungals, which is urgently needed. Minor comments are below.

1. From the description in the materials and methods, APX879 was also examined for efficacy in a murine model of invasive aspergillosis, invasive mucormycosis, and invasive candidiasis but no data are provided. Data should be shown in the supplementary even if there was no substantial efficacy against these infections.
2. The structures are well refined. However, the details of the structural interactions are a bit confusing, even for a structural biologist. It is just so much detail that it is hard to understand what the main point is of all their work. I suggest to re-writing this section.
3. Unclear if the co-crystallization with the new inhibitor (APX879) was done. This should be mentioned.
4. The increase of fungal specificity (APX879 is less immunosuppressive than FK506 in Figure 6) is not translated in the increase of antifungal activity (APX879 is less active than FK506 in Table S3). Please comment.
5. Fig 7B: CFUs should be plotted as "Log10 CFUs/gr organ". Combination 2X clearly shows a decrease in lung CFUs compared to APX879 alone, which is not appreciated in the current graph because the Y axis shows raw numbers. Either change the Y axis into a logarithmic scale or transform the raw CFU data into Log10 and plot the Log10 data using a linear scale on the Y axis.
6. No comparison with FK506-treated mice was provided. This is important because it will show whether APX879 is more efficacious (and less toxic) than its parent compound FK506.
7. APX879 appears to be synergistic (or additive) against *Cryptococcus* in the animal model. However, the fractional inhibitory index (FIC) was not calculated. The authors are encouraged to perform a checkerboard assay and calculate the FIC of APX879 and FK506 when combined with current antifungals (e.g. fluconazole, itraconazole, voriconazole, caspofungin, and amphotericin B). This should reveal that, even if less active than its parent compound FK506 when used alone, APX879 is still synergistic with combined with current antifungals, similar to FK506.

8. More information about the synthesis of the FK506 analogs (e.g. medicinal chemistry), how these derivatives were screened/analyzed, and APX879 was finally selected should be presented.

Reviewer #3 (Remarks to the Author):

In this study, the authors identified amino acid residues critical for binding and inhibition of fungal calcineurin based on NMR, molecular dynamic simulations and mutation analyses. They tried to utilize this information to design and develop a less immunosuppressive FK506 analog with broad-spectrum in vitro antifungal activity and efficacy. Structural analysis is concrete. However, I think that they did not obtain the ideal FK506-related compound. Although the compound (APX879) exhibited reduced mammalian immunosuppressive activity, it also exhibited reduced antifungal activity.

1) Table 3S showed antifungal susceptibility of FK506 and APX878 on different fungi. Although the authors mentioned in the text that APX879 retained in vitro antifungal activity against the major pathogenic fungi (Line 392), it is misleading. This table clearly showed that APX879 is less active than FK506 in pathogenic fungi. Differences in sensitivity between FK506 and APX878 are 32-fold (*A. fumigatus*, AF293 and CEA10), 133-fold (*C. albicans*), 17-fold (*C. neoformans*), 32-fold (*M. circinelloides* f. *lusitanicus*), and 8-fold (*M. circinelloides* f. *circinelloides*). Taking that CD4+ T cells treated with APX879 exhibited a 71-fold reduction in immunosuppressive activity compared to FK506 (Fig. 6C), APX879 is less active both in fungi and mammalian cells. In summary, APX879 is not a non-immunosuppressive CN inhibitor for effective antifungal targeting but is just a less active FK506 derivative.

2) The author used a large amount of APX878 (20 mg/kg) in the Cryptococcal model of murine invasive fungal infection experiment (Fig. 7). This may be due to the fact that APX878 is less active than FK506. It is necessary to ensure that treatment of mice with 20 mg/kg APX878 did not reduce CN activity in mice. The authors should explain the reason why they use this concentration. The authors should also show the control experiments with FK506.

3) It is not clear why the authors only focused on an acetohydrazine substitution of the C22 carboxy. Didn't they use molecular dynamic simulations to select the best analog? Aren't there any possibilities to focus on other analogs?

4) The authors should show more quantitative data and show the number of replicates in each experiment. For example, Table S4 only represents S (sensitive) and R (resistant) to describe sensitivity to the compounds. It is too qualitative.

Nature Communications: Reviewers' Comments

We appreciate the reviewers for their helpful comments and suggestions for our manuscript. We have now addressed the concerns raised by the reviewers in the revised version of the manuscript with additional experimentation and text modifications. All modifications to the text in response to the review are in red color. We think these new inclusions have significantly improved the overall manuscript and hope that our work is now suitable for consideration of publication in *Nature Communications*.

Point by Point Response

Reviewer #1:

The manuscript details experimental (X-ray crystallography, NMR spectroscopy, site directed mutagenesis, in vitro activity, in vivo therapeutic efficacy in an infection model) and theoretical (model building, docking and molecular dynamics simulations) investigations towards the development of a less immunosuppressive FK506 analog, APX879, as novel effective antifungal therapeutics.

This work is an extension on previously reported work by the authors [most recently 'Structures of pathogenic fungal FKBP12s reveal possible self-catalysis function', (2016) *MBio* 7 e00492-e00416], however here the authors report the first crystal structures of the CN ternary complexes with FKBP12-FK506 from four highly clinical relevant human fungal pathogens.

This is an impressive structure/activity/functional study where some of the structural elements of the ternary complexes were recognized as essential for binding and inhibiting specificity of fungal versus mammals CN interactions, namely the key residue Phe88 of FKBP12 which is not conserved in mammalian FKBP12 (His88).

The manuscript is clearly written and read generally well, the experimental and theoretical methodology is appropriate and the results appear to be valid and are well presented, founded on a thorough structural and in vitro / in vivo biological data analysis.

Response: We thank the reviewer for these positive comments.

However, I have two major concerns:

i) The development of APX879, a FK506 analog, based on the structural differences of the human and fungal ternary CN complexes, is an interesting and encouraging finding in the search of more potent and selective anti-fungal. However, APX879 exhibits, on activated naïve CD4+T cells, just a 71-fold reduction in immunosuppressive activity compared to FK506 and it shows a limited therapeutic efficacy in the tested murine fungal infection models.

Response: We agree that APX879 has limited efficacy in the tested infection models, but as the reviewer positively indicated, this compound is interesting as its design is based on structural differences between human and fungal complexes. This is our first lead compound obtained after screening several analogs with different substitutions at the C22 position on FK506 (now shown in Table S3). We are currently working on designing more potent analogs to address this important issue. We think that the structural knowledge

gained from the different CN ternary complexes is critical to advancing the design of newer compounds with lower or no immunosuppressive activity in the future.

APX879 exhibited 71-fold reduction in immunosuppressive activity and a 17-fold reduced efficacy in *Cryptococcus* antifungal activity, which is approximately 4-fold improvement in the therapeutic balance between immunosuppression and antifungal activity. To further address the reduction of *in vivo* immunosuppressive activity of APX879 compared to FK506, we have also now measured the immunosuppressive state of mice treated with vehicle, APX879 (20 mg/kg), or FK506 (5 mg/kg) as a positive control. Uninfected, healthy mice were immunized with NP-OVA (Day 0), to stimulate a T-cell dependent Germinal Center (GC) B cell proliferation. Animals were treated with vehicle, FK506, or APX879 prior to immunization (Day -1) and continued until Day 7 with a daily single dose of vehicle, APX879, or FK506. The lymph nodes of each individual animal were harvested on day 7 and relative abundance of T helper cells and GC B cells was measured. As shown below this experiment indicated that APX879 exhibited significantly reduced *in vivo* immunosuppressive activity in comparison to FK506. This new data is included in Figure 6 (Panels D and E) and explanation is included in Lines 384-389.

Moreover, APX879 also showed improved efficacy over the parent compound FK506, which was acutely toxic at 20 mg/kg in our *Cryptococcosis* model. Lower doses of FK506 have been previously shown to be ineffective in a murine systemic *cryptococcosis* model (at 3 mg/kg) and the wax moth *Galleria mellonella* model (at 5 mg/kg) (Lee et al. AAC. 2018). These new data are included in Fig.7 (Panel A) and explanation is included in Lines 405-416.

ii) The general significance and interest of this manuscript IMHO, it is too medicinal chemistry focused, for the diverse readership of Nature Communications. Overall, I strongly recommend the authors to submit their manuscript to a more specialized chemical biology or medicinal chemistry focused journal, i.e. Cell Chemical Biology, ACS Chemical Biology, and ACS Journal of Medicinal Chemistry.

Response: We present CN ternary complex structural data from four different human fungal pathogens in comparison to the mammalian structure to provide a broader scope of the work, including relevance to the structural biology and microbiology fields. We also think the genetic work performed based on structural biology and substantiated by molecular dynamic simulations provides a more general proof of concept. The final section of the manuscript focuses on the rational design of an inhibitor based on the structural differences, and includes antifungal susceptibility testing and testing in multiple animal models of infection, which is a less chemical aspect of the work and may not be suitable for a purely medicinal chemistry journal.

Minor Comments:

1. The Abstract should directly mention the results for APX879 for the sake of clarity.

Response: We have included this information in the Abstract in Lines 54-56.

2. The authors are kindly requested to adopt throughout the manuscript a “unifying” numbering scheme for the protein residues. It should be consistent with those in the deposited related PDB files. For instance, in the PDB entry 5HWB (A. fumigatus FKBP12) the numbering of selected residues is: Asp38, Arg43, Phe88 and Ile92, whereas in the PDB entries 1TCO (B. taurus CN-FKBP12-FK506 ternary complex), 2PPN (human FKBP12) and 1FKJ (human FKBP12-FK506 complex) the numbering is Asp37, Arg42, His87 and Ile90. In the PDB entry 5b8i (C. immitis CN-FKBP12-FK506 ternary complex) the numbering instead is Asp56, Arg61, Phe105 and Leu108.

Response: We thank the reviewer for this suggestion and this is something that we also considered before submission. However, as shown below, due to sequence gaps/insertions between the different species discussed, we struggled to find a common numbering convention. As we show in the alignments below for the human, *Aspergillus*, *Candida*, *Cryptococcus*, and *Coccidioides* Calcineurin A, Calcineurin B and FKBP12 sequences, the yellow highlighted regions lead to sequence numbering differences between not only human but also between different fungal species. While it is conceivable to align the numbering for the Calcineurin A and B sequences as most of the sequence gaps/insertions are located at the extreme N- and C-termini, for FKBP12, the gaps/insertions are located throughout the sequence, making it challenging to find a common register. While the AfCnA sequence is shifted +24, AfCnB sequence is shifted -23, and AfFKBP12 is shifted -1, it would also lead to differences in numbering for other fungi. This would result in numerous changes in the text and supplementary tables, and also be confusing with respect to the fungal sequences numbering. Considering the potential for considerable confusion, we prefer to retain the sequence numbering presented.

Calcineurin A

	10 20 30 40 50 60
Aspergillus	-----GS-----EDGIQV
Coccidioides	-----MDGSKIVRAVTEKK--PVPE-IDFTLHTEMEDGTQV
Cryptococcus	-----MASPATQTANAIAAINNRSNLVIPE-IDFTQHQLENGELV
Human	-----MSEPK-----AIDPKL
Candida	MSGNTVQRNTEQINNALNAIQHRRTTIGNESNTNTNQRQILQNPTGRDYTIYVTDGGEKY
	70 80 90 100 110 120
Aspergillus	STLERVIKEVQAPALSTPTDEMFWSPEDPSKPNLQFLKQHFYREGRLTEEQALWIIHAGT
Coccidioides	STLERVCKEVQAPAFHTPTNEQFWSPVDPSPKNLAFKQHFYREGRLTEEQALWIIHAGT
Cryptococcus	STTERVIKDVQAPAMYVPTDDQFWQKVDKTKPDIAFLKNHFYREGRLTEEQALYILEKGG
Human	STTDRVVKAVPFPFSHRLTAKEVFD--NDGKPRVDILKAHLMKEGRLEESVALRIITEGA
Candida	STVERAVKSDVPATFKPKDEQVFFY--NGKPNHQFLKQHFIIHEGRLHEHQAIQILKQAT ** : * : * : * : : ** : ** * : : ** * * * : * : * : .
	130 140 150 160 170 180
Aspergillus	QILRSEPNLLEMDAPITVCGDVHGQYYDLMKLFEVGGDPSETRYLFLGDYVDRGYFSIEC
Coccidioides	ELLRAEPNLLEMDAPITVCGDVHGQYYDLMKLFEVGGDPAEIRYLFLGDYVDRGYFSIEC
Cryptococcus	ELLRSEPNLLEVDAPITVCGDIIHGQYYDLMKLFEVGGNPADIRYLFLGDYVDRGYFSIEC
Human	SILRQEKNLDDIDAPVTVCGDIHQGFDFMLKLFVGGSPANTRYLFLGDYVDRGYFSIEC
Candida	HLLSKEPNLLSVPAPVTICGDVHGQYYDLMKLFEVGGDPASTKYLFLGDYVDRGSFSIEC : * * * * : * : * : * : * : * : * : * : * : * : * : * : * : * : * : *
	190 200 210 220 230 240
Aspergillus	VLYLWALKIWPNSLWLRLGNHECRHLDYFTFKLECKHKYSERIYEACIESFCALPLAA
Coccidioides	VLYLWALKIWPNTLWLRLGNHECRHLDYFTFKLECKHKYSEKVVYDACMESFCALPLAA
Cryptococcus	VLYLWSLKMWPDTLFLRLGNHECRHLDYFTFKLECKHKYSETVYNACMESFCNLPLAA
Human	VLYLWALKILYPKTLFLRLGNHECRHLTEYFTFKQECKIKYSERVYDACMDAFDCLPLAA
Candida	LLYLYSLKINYPDTFWMLRGNHECRHLTEYFTFKNECLHKYSEELYEECLVSNALPLAA : * * : * * : * : * : * : * : * : * : * : * : * : * : * : * : * : * : *
	250 260 270 280 290 300
Aspergillus	VMNQFLCIHGGLSPHELHLEDIKSIDRFREPPTHGLMCDILWADPLEDFGQEKTGDYFV
Coccidioides	IMNQFLCIHGGLSPHELHLEDIKSIDRFREPPTHGLMCDILWADPLEDFGTEKTGEYFV
Cryptococcus	VMNQFLCIHGGLSPHELHLDLRSINRFREPPTQGLMCDILWADPLEDFGSEKTENENFL
Human	LMNQFLCVHGGLSPEINTLDDIRKLDRFKPPAYGPMCDILWSDPLEDFGNEKTEHEFT
Candida	IMNEQFCVHGGLSPQLTSLDSLRLKLRFRPPTKGLMCDLLWADPIEYDDNLDQYV : * * * * : * : * : * : * : * : * : * : * : * : * : * : * : * : * : *
	310 320 330 340 350 360
Aspergillus	HNSVRGCSYFFSYPAACAFLEKNNLLSIIIRAHEAQDAGYRMYRKTITGFPSVMTIFSAP
Coccidioides	HNNVRGCSFFFSYPAACAFLEKNNLLSIIIRAHEAQDAGYRMYRKTITGFPSVMTIFSAP
Cryptococcus	HNVVRGCSYFFTYNAACQFLERNLLSIIIRAHEAQDAGYRMYRKTITGFPSVMTIFSAP
Human	HNTVRGCSYFYSPVAVCEFLQHNNLLSILRAHEAQDAGYRMYRKSQITGFPSLTIIFSAP
Candida	TNVVRGCSFAFTYKAACKFLDRTKLLSIIIRAHEAQNAGYRMYRKTITMGFPSLLTFMSAP * * * * : * : * : * : * : * : * : * : * : * : * : * : * : * : * : *
	370 380 390 400 410 420
Aspergillus	NYLDVYNNKAAVLKYENNVNIRQFNCTPHPYWLPNFMVFTWVSLPFVGEKIDMLIAIL
Coccidioides	NYLDVYNNKAAVLKYENNVNIRQFNCTPHPYWLPNFMVFTWVSLPFVGEKIDMLIAIL
Cryptococcus	NYLDVYNNKAAVLKYESNVMNIRQFNCTPHPYWLPNFMVFTWVSLPFVGEKIDMLIAIL
Human	NYLDVYNNKAAVLKYENNVNIRQFNCSHPHYWLPNFMVFTWVSLPFVGEKVTEMLVNVL
Candida	NYLDSYNNKAAVLKYENNVNIRQFNASHPHYWLPNFMVFTWVSLPFVGEKVTDMLVSL * * * * : * : * : * : * : * : * : * : * : * : * : * : * : * : * : *
Aspergillus	NTC
Coccidioides	NTC
Cryptococcus	NCC
Human	NIC
Candida	NVC
	* *

Calcineurin B

	10	20	30	40	50	60
Aspergillus	MEQPSEPNNAAMYDARRRRASVGT	SQLLDNIVSASNFDRDEVDRLRKR	FMKLDKDS	SGTI		
Coccidioides	MTE--QDNNAESYDATKRQQTSSSS	SQVLNDIVSGSNFDHEEVDRLWKR	FMKLDKDR	KSGTI		
Cryptococcus	-----MGAAES	SMFNSLEKNSN	FSGPELMRLK	KRFMKLDKDG	SGSI	
Candida	-----MGANAS	ILLDGFIEDT	NFSIEEIDRLR	KRFMKLDKDG	SGQI	
Human	-----MGNEAS	YPLEMCSHFDAE	EIKRLGKR	FKKLDL	NSGSL	
			*	:	::*	*: ** *** ** * ** :
	70	80	90	100	110	120
Aspergillus	DRDEFSLSPQVSSNPLATRMIAI	FDEDEGGDVDFQEFV	SGLSAFSSKGNKEE	KLRF	FAFKV	
Coccidioides	ERDEFSLSPQVSSNPLSTRMIAI	FDEDEGGDVDFQEFV	SGLSAFSSKGNKEE	KLRF	FAFKV	
Cryptococcus	DKDEFLQIPQIANNPLAHRMIAI	FDEDEGSGTVDFQEFV	GGLSAFSSKGG	RDEKLRF	FAFKV	
Candida	DKQEFSLIPGISNPLATRLMDV	FDKDGDSIDFEEFITGL	SAFSGKSDN	LNLKLR	FAFNI	
Human	SVVEEFMSLPELQQNPLVQRVIDI	FDTDGNGEVDFKEF	IEGVSQF	SVKGDKE	QKLRF	AFRI
			.	::**	::**	::**
	130	140	150	160	170	180
Aspergillus	YDIDRDGYISNGELFIVLKM	MVGNLKD	VQLQQIVDKT	IMEADKDR	DGKIS	FEEFT
Coccidioides	YDIDRDGFTISNGELFIVLKM	MVGNLKD	MQLQQIVDKT	IMEADLDG	GRIS	FEEFTR
Cryptococcus	YDMDRDGYISNGELYLVLK	QMVGNLKD	QQLQQIVDKT	IMEADKDG	DKLS	FEEFTQ
Candida	YDIDRDGYISNGELFIVM	KMVGNLKD	EELQQIVDKT	IMEADLDG	DKLN	FEEFKNA
Human	YDMDRDGYISNGELFQVLK	MVGNLKD	TQLQQIVDKT	IINADK	DGGRIS	FEEFC
			::	::**	::**	::**
	190					
Aspergillus	NTDVSLSM	TLSMF				
Coccidioides	NTDVSM	SLDQF				
Cryptococcus	STDIVK	QMTLEDLF				
Candida	TDTIANT	LTLNMF				
Human	GLDIHKK	MVVDV				
			:	:::		

FKBP12

	10	20	30	40	50	60
Candida	MSEELPQIEIVQEGDNTTFAKPG	DVTVIHYDGKLTN	-----GKE	FDS	SRKRGK	PFT
Cryptococcus	MG---VTVENISAGDGKTFPQP	GDVNTIHYVGTLLD	-----GSK	FDS	SRDRG	TPFV
Aspergillus	MG---VTKELKSPGNVDFPKG	DFVTIHYTGRLTD	-----GSK	FDS	SVDRNE	PFQ
Coccidioides	MG---VTKKILKEGNVDPVK	GDDIVMNYRGCLYDSSK	PSEHFMGR	KFDS	TEERGE	FK
Human	MG---VQVETISPGDGRTPPKR	GQTCVHYTGLED	-----GKK	FDS	SRDRNK	PFK
			*	:	::*	*: ** ** * * * * * * * * * * :
	70	80	90	100	110	120
Candida	CTVGVGQVIRKWDISL	TNNYKGGANL	PKISKGT	KAILTI	PPNLAYG	PRGIPPI
Cryptococcus	CRIGQGQVIRKWD	-----EG	---VPQL	SVGQKAN	LICTPDY	AYGARGFP
Aspergillus	TQIGTGRVIRKWD	-----EG	---VPQMS	LGEKAVL	TITPDY	GYGARGFP
Coccidioides	TKIGIGVIRKWD	-----EA	---VLQMS	LGEKSI	LITITDDY	AYGARGFP
Human	EMLGQEVIRKWE	-----EG	---VAQMS	VGQRAK	LITISPDY	AYGATGHP
			::*	::**	::**	::**
	130					
Candida	LVFEV	ELLGVNGQ				
Cryptococcus	LKFEV	ELLKVN				
Aspergillus	LIFEV	ELLGINN	KRA			
Coccidioides	LVFEV	ELKGIN	SKRA			
Human	LVFDV	ELLKLE				
			*	::**	::	

3. Page 6, line 145: “are close to FKBP12” should read “are close to FK506”

Response: We have corrected this now on Line 137.

4. Page 31, Table 1: In Table 1, from a methodological point of view, the Pearson correlation coefficients CC* and the CC1/2 values instead of the R-merge residual should be provided. Reporting the indicator R-merge is quite a controversial issue. See: Karplus P.A, Diederichs K. "Assessing and maximizing data quality in macromolecular

crystallography. *Curr Opin Struct Biol.* (2015) 34, 60-68; Diederichs K, Karplus PA. “Better models by discarding data?” *Acta Crystallogr D Biol Crystallogr.* (2013) 69, 1215-1222; Diederichs K., Karplus P.A. “Linking crystallographic model and data quality” *Science* (2012) 336, 1030-1033; Karplus P.A., Diederichs K. “Improved R-factors for diffraction data analysis in macromolecular crystallography” *Nat Struct Biol* (1997) 4, 269-275 and Erratum in *Nat Struct Biol* (1997) 4, 592.

Response: We thank the reviewer for citing these important publications and we have now replaced R-merge with CC* and CC1/2 values in Table 1.

PDB ID	6BLT	6BLU	5B8I	6BLV
Species	C. albicans	A. fumigatus	C. immitis	C. neoformans
Data Collection				
Space Group	P2₁2₁2₁	P2₁	P2₁2₁2₁	P2₁2₁2₁
Cell dimensions				
a, b, c (Å)	62.47, 142.85, 175.61	59.24, 94.46, 69.83	94.67, 154.63, 64.75	118.72, 120.70, 134.
α, β, γ (°)	90.00, 90.00, 90.00	90.00, 109.28, 90.00	90.00, 90.00, 90.00	90.00, 90.0, 90.00
Resolution (Å)	2.55 (2.62-2.55)	2.50 (2.56-2.50)	1.85 (1.90-1.85)	3.30 (3.39-3.30)
CC(1/2)	99.8 (82.6)	99.6 (72.8)	99.8 (92.3)	99.2 (91.8)
CC*	99.9 (95.3)	99.9 (91.7)	99.9 (98.2)	99.8 (98.0)
I / σ (I)	16.87 (3.37)	12.00 (2.09)	15.70 (4.20)	8.62 (3.34)
Completeness (%)	98.4 (99.2)	99.7 (99.9)	100.0 (100.0)	99.9 (99.9)
Redundancy	4.4 (4.5)	3.8 (3.9)	7.4 (7.5)	6.2 (6.3)
Refinement				
Resolution (Å)	2.55	2.5	1.85	3.3
No. Reflections	51,429	25,206	81,827	29,795
R _{work} / R _{free} overall	18.2 / 22.9	20.0 / 23.2	14.5 / 17.2	19.9 / 24.3
Molprobability				
Ramachandran				
Favored (%)	96.43	96.07	96.89	94.84
Outliers (%)	0.17	0.00	0.16	0.31
Rotamers				
Favored (%)	92.62	96.82	96.32	90.04
Outliers (%)	2.81	0.64	1.23	1.93
MolProbity score	1.83 (98th)	1.28 (100th)	1.35 (98th)	1.78 (100th)
No. Atoms				
Protein	9429	4643	5260	10132
CnA	5749	2879	2974	6028
CnB	2046	875	1374	2510
FKBP12	1634	889	912	1594
Ligand	148	75	126	141
FK506	114	57	57	114
Water	281	103	687	9
ADP (Å²)				
Protein	52.66	47.01	25.68	40.20
CnA	42.31	43.43	20.04	36.21
CnB	70.32	62.50	29.89	49.00
FKBP12	66.97	43.38	37.76	41.42
Ligand	47.41	41.17	27.90	32.89
FK506	43.20	36.08	19.71	27.03
Water	37.95	39.65	37.62	20.99
r.m.s. deviations				
Bond lengths (Å)	0.008	0.002	0.02	0.005
Bond angles (°)	0.916	0.597	1.641	1.03

The authors should distinctly specify in Table 1, the N° of atoms and ADP (Å²) values for CN-A, CNB, FKBP12 and FK506 respectively. Please specify the chemical nature of the “Ligand”.

Response: We have separated out the number of atoms and ADP (Å²) values for CnA, CnB, FKBP12 and FK506 in Table 1. The “Ligand” is defined as all non-water, non-protein atoms, including FK506, but we have now broken out FK506 specifically from the larger group of “Ligands.”

In Table 1 for each of the structures, should also be included the values representative of the Ramachandran plots (Favored / Outliers (%)) as well as those of the MolProbity validation analysis, i.e. Rotamer outliers (%) and the MolProbity scores.

Response: We thank the reviewer for this recommendation and we have updated the Table 1 with values for Ramachandran plots (Favored / Outliers (%)), Rotamers (Favored / Outliers (%)) and the MolProbity score.

5. Page 34, line 833: “of approximately 2.5e6 cells/ml” should read “of approximately 2.5·10⁶ cell/mL”

Response: We have corrected this now on Line 850.

6. Page 41, line 1003: The authors indicate PDB entries 6BLS and 6BLU. In Table 1, the crystallography and structure refinement statistics for the 6BLS PDB entry are absent. Moreover the authors should clearly state that the 6BLS PDB entry is related to the crystal structure of the ternary complex of *A. fumigatus* Calcineurin A, Calcineurin B, the P90G FKBP12 mutant and FK506.

Response: Thanks for pointing this out. The PDB entry for 6BLS was removed from the “Discussion” in this manuscript prior to submission. We inadvertently left in this reference to 6BLS. It has been deleted.

Page 42, line 1004: “were grown in Proplex condition” should read “were grown in Proplex (Molecular Dimensions) condition”.

Response: We have corrected this now on Line 1021-1022.

Page 42, line 1015: “screen matrix screen (Rigaku reagents) based on Morpheus II” should read “screen matrix (Molecular Dimensions reagents) based on Morpheus II”.

Response: We have corrected this now on Line 1032.

Page 52, line 1249: The authors are kindly request to clarify the meaning of “during rigid body EM was set to false”.

Response: We have clarified this point. Changed: "sampling of 180 degree rotated solutions during rigid body EM was set to false" To: "sampling of 180 degree rotated solutions during iteration 0 (rigid docking) was set to false" now on Line 1271.

Figure 2, Legend: In the last sentence, the authors should clearly state the PDB entry 5HWB is related to the WT *A. fumigatus* FKBP12 crystal structure while the PDB entry

5HWC is related to the crystal structure of the *A. fumigatus* FKBP12 mutant P90G in complex with FK506.

Response: We thank the reviewer for pointing this out. Actually, this should be read as 5HWB. The 5HWC refers to our earlier P90G mutation structure.

Figure 6, Panel C: The IC50 values of FK506 and APX879 are given without any error estimates.

Response: Thanks for pointing this out. The plot is showing 95% confidence interval for the 4-parameter curve fit, error bars for individual points represent the SE. The IC50 estimates do have a confidence associated with them, these are the 95% CI's for the estimates:

FK506: 0.1692 to 0.2122

APX879: 10.74 to 16.85

These are asymmetrical (i.e. not described by +/-) and called profile likelihood asymmetric confidence intervals. In essence, the IC50 is iteratively changed from its best-fit value until the p-value is no longer significant. The upper and lower value at which the p-value is exactly 0.05 are reported as the upper and lower bounds of the 95% confidence interval.

We have briefly included this explanation in the Figure 6 legend for clarity.

Figure S1, Legend: Panel (F), the PDB entry PDB 1TCO is not human FKBP12 but rather *B. taurus* CN-FKBP12-FK506 ternary complex.

Response: This has been corrected by replacing human with bovine in Figure S1.

Figure S3: The entities and the related units (X- and Y-axes) have been omitted in all the plots (Panels A-E).

Response: We have corrected this in the Figure S3.

Figure S8: The authors should indicate in the legend that the molecular surface is colored according to the calculated electrostatic potential contoured from $-XX$ kT/e (red) to $+YY$ kT/e (blue). The same apply for the hydrophobic surface plot.

Response: We have corrected this in Figure S8 (now Figure S11) on Lines 1461-1465.

Reviewer #2:

The paper by Juvvadi et al. reports the crystal structure of calcineurin catalytic and regulatory subunit complexed with FK506 and FKBP12 from the most common human fungal pathogens. Calcineurin is required for fungal growth and inhibitors of this complex, such as FK506 or Cyclosporin A, are fungicidal. However, these inhibitors are potent immunosuppressive in mammalian cells, limiting their use in the clinic against fungal infections. Using a structural biology approach, the authors found key residue(s) in the fungal enzyme complex not conserved in the human counterpart. This difference was used to design and synthesize derivatives of FK506, such as APX879, with higher affinity to the fungal complex. As a result, APX879 is much less immunosuppressive than FK506. Importantly, APX879 maintains the broad spectrum of antifungal activity of the parent compound and it is also efficacious in a murine model of invasive fungal infection.

I found the studies of this paper convincing, well-illustrated and thoroughly discussed. I also found the experimental design well thought out with appropriate controls in place. I also think these studies are highly significant, and they will have a dramatic impact in the research and development of new antifungals, which is urgently needed.

Response: We thank the reviewer for these positive comments.

Minor comments:

1. From the description in the materials and methods, APX879 was also examined for efficacy in a murine model of invasive aspergillosis, invasive mucormycosis, and invasive candidiasis but no data are provided. Data should be shown in the supplementary even if there was no substantial efficacy against these infections.

Response: These data as shown below are now provided in the revised manuscripts Supplemental data, as Supplemental Figure S10. We have included a sentence to clarify this point on Lines 403-405).

2. The structures are well refined. However, the details of the structural interactions are a bit confusing, even for a structural biologist. It is just so much detail that it is hard to understand what the main point is of all their work. I suggest to re-writing this section.

Response: We thank the reviewer for this thoughtful comment and we have revised the manuscript to clarify and reduce structural discussions as it pertains to elements that are not central to the theme of the manuscript. We have included some important

explanations pertaining to the structures in the respective legends for figures (Figure 1, Figure S1-S2 and Figure S4) to make it clear to the reader.

3. Unclear if the co-crystallization with the new inhibitor (APX879) was done. This should be mentioned.

Response: We have included a statement in the “Discussion” on Lines 493-496, noting that the ternary complex can be readily obtained through recombinant protein expression in the presence of APX879; however, after two attempts, we have been unable to crystallize the complex under similar conditions that led to the FK506 complex crystal structure.

4. The increase of fungal specificity (APX879 is less immunosuppressive than FK506 in Figure 6) is not translated in the increase of antifungal activity (APX879 is less active than FK506 in Table S3). Please comment.

Response: We agree that the significantly lowered immunosuppressive activity of APX879 in comparison to FK506 is not directly translated to an increase in antifungal activity of APX879. The purpose of this design is to maintain antifungal activity while lowering the immunosuppressive activity to a point where the therapeutic window is reached. In this regard, an important point to consider is that APX879 exhibited 71-fold reduction in immunosuppressive activity and a 17-fold reduced efficacy in *Cryptococcus* antifungal activity, which is approximately 4-fold improvement in the therapeutic window. APX879 is our first lead compound after screening several analogs with different substitutions at the C22 position (now shown in Table S3), and we are now working on designing more potent analogs to improve the antifungal activity. For clarification, we have included explanations related to this point on Lines 403-416 and 513-525.

5. Fig 7B: CFUs should be plotted as “Log10 CFUs/gr organ”. Combination 2X clearly shows a decrease in lung CFUs compared to APX879 alone, which is not appreciated in the current graph because the Y axis shows raw numbers. Either change the Y axis into a logarithmic scale or transform the raw CFU data into Log10 and plot the Log10 data using a linear scale on the Y axis.

Response: We thank the reviewer for pointing this out. As shown below the CFUs are plotted as Log10 values in the Figure 7B (now Figure 7C).

6. No comparison with FK506-treated mice was provided. This is important because it will show whether APX879 is more efficacious (and less toxic) than its parent compound FK506.

Response: We have addressed this important concern. To determine the efficacy of FK506 treatment in our cryptococcosis model, we treated *C. neoformans* infected animals

with vehicle (PBS) or 20 mg/kg of FK506. In this model, we found that 20 mg/kg of FK506 was acutely toxic. All FK506 treated animals died by day 3, and 9 out of 10 mice died within 48 hours following FK506 treatment. The median survival of FK506-treated animals was 1.5 days while the median survival time for vehicle treated animals was 23 days. We have now included this data in Figure 7 (Panel A) and clarified this point on Lines 405-416). Lower doses of FK506 have been previously shown to be ineffective in a murine systemic cryptococcosis model (at 3 mg/kg) and the wax moth *Galleria mellonella* model (at 5 mg/kg) (Lee et al. AAC. 2018).

7. APX879 appears to be synergistic (or additive) against *Cryptococcus* in the animal model. However, the fractional inhibitory index (FIC) was not calculated. The authors are encouraged to perform a checkerboard assay and calculate the FIC of APX879 and FK506 when combined with current antifungals (e.g. fluconazole, itraconazole, voriconazole, caspofungin, and amphotericin B). This should reveal that, even if less active than its parent compound FK506 when used alone, APX879 is still synergistic with combined with current antifungals, similar to FK506.

Response: We appreciate this reviewer’s suggestion. FK506 has been shown to have synergistic effect with fluconazole in *Cryptococcus*. We have performed checkerboard assays for APX879 and FK506 in combination with clinically relevant antifungals against the different fungal species. The FIC index data is included in Table S6. We found that, similar to FK506, APX879 is also synergistic with amphotericin B or Ambisome, and the echinocandin Caspofungin in other fungi. We have clarified this point on Lines 377-379.

Strain	Calcineurin Inhibitor (A)	Antifungal (B)	MIC (A)	MIC (B)	FIC Index
Aspergillus fumigatus (CEA10)	FK506	Ambisome	0.0312	1	0.187
	APX879	Ambisome	1	1	0.187
	FK506	Caspofungin	0.0312	1	0.218
	APX879	Caspofungin	1	1	0.375
	FK506	Voriconazole	0.0312	0.25	≤2
	APX879	Voriconazole	1	0.25	≤2
Candida albicans (SC5314)	FK506	Amphotericin B	0.06	0.25	0.266
	APX879	Amphotericin B	8	0.25	0.254
	FK506	Caspofungin	0.06	0.25	0.375
	APX879	Caspofungin	8	0.25	0.375
Cryptococcus neoformans (H99)	FK506	Amphotericin B	0.06	0.25	0.265
	APX879	Amphotericin B	1	0.25	0.258
Mucor circinelloides f. circinelloides	FK506	Ambisome	0.25	0.125	0.75
	APX879	Ambisome	2	0.125	0.625
	FK506	Isavuconazole	0.25	8	≤2
	APX879	Isavuconazole	2	8	≤2

8. More information about the synthesis of the FK506 analogs (e.g. medicinal chemistry), how these derivatives were screened/analyzed, and APX879 was finally selected should be presented.

Response: As suggested, we have now included a flow chart detailing the procedures followed in screening the various FK506 analogs (Figure S8). A new table (Table S3) showing 16 FK506 analogs synthesized through substitution at the C22 position of FK506 is also included. References pertaining to the synthesis and screening procedures are added to the list of references.

Reviewer #3:

In this study, the authors identified amino acid residues critical for binding and inhibition of fungal calcineurin based on NMR, molecular dynamic simulations and mutation analyses. They tried to utilize this information to design and develop a less immunosuppressive FK506 analog with broad-spectrum *in vitro* antifungal activity and efficacy. Structural analysis is concrete. However, I think that they did not obtain the ideal FK506-related compound. Although the compound (APX879) exhibited reduced mammalian immunosuppressive activity, it also exhibited reduced antifungal activity.

1) Table 3S showed antifungal susceptibility of FK506 and APX878 on different fungi. Although the authors mentioned in the text that APX879 retained *in vitro* antifungal activity against the major pathogenic fungi (Line 392), it is misleading. This table clearly showed that APX879 is less active than FK506 in pathogenic fungi. Differences in sensitivity between FK506 and APX878 are 32-fold (*A. fumigatus*, AF293 and CEA10), 133-fold (*C. albicans*), 17-fold (*C. neoformans*), 32-fold (*M. circinelloides* f. *lusitanicus*), and 8-fold (*M. circinelloides* f. *circinelloides*). Taking that CD4+ T cells treated with APX879 exhibited a 71-fold reduction in immunosuppressive activity compared to FK506 (Fig. 6C), APX879 is less active both in fungi and mammalian cells. In summary, APX879 is not a non-immunosuppressive CN inhibitor for effective antifungal targeting but is just a less active FK506 derivative.

Response: We understand the reviewer's concern and as already mentioned in the manuscript we agree that APX879 is not completely non-immunosuppressive but exhibited less immunosuppression when compared to its parent compound FK506. Despite a decrease in antifungal activity compared to FK506, APX879 maintained broad spectrum *in vitro* antifungal activity against the major human fungal pathogens we tested. An important point to consider is that APX879 exhibited a 71-fold reduction in immunosuppressive activity and a 17-fold reduced efficacy in *Cryptococcus* antifungal activity, which is approximately a 4-fold improvement in the therapeutic window. APX879 is our first lead compound after screening several analogs with different substitutions at the C22 position (now shown in Table S3), and we are now working on designing more potent analogs to improve the antifungal activity.

Although there is a clear reduction in antifungal activity against all of the fungal pathogens tested in this study, the reduction in immunosuppressive activity of APX879 compared to FK506 is larger than the reduction in antifungal activity in most cases and this contributed to the efficacy of APX879 in our cryptococcosis model. To further address the reduction of *in vivo* immunosuppressive activity of APX879 compared to FK506, we have also now measured the immunosuppressive state of mice treated with vehicle, APX879 (20 mg/kg), or FK506 (5 mg/kg) as a positive control. Uninfected, healthy mice were immunized with NP-OVA (Day 0), to stimulate a T-cell dependent Germinal Center (GC) B cell proliferation. Animals were treated with vehicle, FK506, or APX879 prior to immunization (Day -1) and continued until Day 7 with a daily single dose of vehicle, APX879, or FK506. The lymph nodes of each individual animal were harvested on day 7 and relative abundance of T helper cells and GC B cells was measured. As shown below this experiment indicated that APX879 exhibited significantly reduced *in vivo* immunosuppressive activity in comparison to FK506. This new data is included in Figure 6 (Panels D and E) and explanation is included in Lines 384-389.

To further substantiate the immunosuppressive activity difference between FK506 and APX879 we have now included new data on the efficacy of FK506 treatment in our murine model of cryptococcosis, which indicated that 20 mg/kg of FK506 was acutely toxic and resulted in 100% mortality by day 3, while 20 mg/kg of APX879 was efficacious in extending survival (Figure 7A). In relation to this, we have now included explanations on Lines. 377-379, 403-416, 513-525 to emphasize these points.

2) The author used a large amount of APX879 (20 mg/kg) in the Cryptococcal model of murine invasive fungal infection experiment (Fig. 7). This may be due to the fact that APX879 is less active than FK506. It is necessary to ensure that treatment of mice with 20 mg/kg APX879 did not reduce CN activity in mice. The authors should explain the reason why they use this concentration. The authors should also show the control experiments with FK506.

Response: The dose of 20 mg/kg of APX879 was selected following an MTD study (data not shown) that showed no weight loss or signs of toxicity in healthy animals treated with 20 mg/kg of APX879 daily for 5 days. As suggested by the reviewer we have performed control experiment with FK506, which indicated that 20 mg/kg of FK506 was acutely toxic and resulted in 100% mortality by day 3 (Figure 7A). A recent study also reported the ineffectiveness of lower dose of FK506 (3 mg/kg) in systemic cryptococcosis model.

3) It is not clear why the authors only focused on an acetohydrazine substitution of the C22 carboxy. Didn't they use molecular dynamic simulations to select the best analog? Aren't there any possibilities to focus on other analogs?

Response: Based on our *A. fumigatus* CN complex structural model, the proximity of Phe88 and Val91 residues to the C22 carbonyl of FK506 in the bound conformation was an ideal starting point for semi-synthetic modifications of the complex macrolide. We hypothesized that small derivatives of the C22 carbonyl group would sterically clash to a

greater extent with His88 of the mammalian complex when compared to Phe88 in the fungal complex. To test this, a series of oximes and alkyl- and hydrazide hydrazones were synthesized. We have now included this explanation on Lines 362-368. A flow chart detailing the procedures followed in screening of the various FK506 analogs is also included as Figure S8. As part of the development of a suitable FK506 analog, we synthesized several compounds with different substitutions at the C22 position and only the C22-acylhydrazones performed the best in our screening. A table (Table S3) showing 16 FK506 analogs synthesized through substitution at the C22 position of FK506 is also now included. References pertaining to the synthesis and screening procedures are added to the list of references.

Additionally we utilized MD simulations to select for the best target residue as shown in Figure 5. The most contacts between FKBP12 and CNA occur in residues 88-91 and only at residue 88 there is a drastic change in amino acid type. The F88 residue is possibly the most appropriate residue to target for inhibitor design.

4) The authors should show more quantitative data and show the number of replicates in each experiment. For example, Table S4 only represents S (sensitive) and R (resistant) to describe sensitivity to the compounds. It is too qualitative.

Response: The main point of the Table S4 (now Table S5) was to show that APX879 is an FK506 analog and does target CN via FKBP12 binding in the different fungi. The WT strains of all the fungal species are in fact susceptible/sensitive to both FK506 and APX879, as the inhibitors bind to FKBP12 and inhibit calcineurin. In the absence of FKBP12, the respective deletion strains show complete resistance to FK506 and APX879. These data are not fully quantitative as we are observing a binary presence or absence of a zone of inhibition on solid agar plate assays (as indicated in the methods). To clarify the table we have now modified as R=Resistant (no zone of inhibition present), S=Sensitive (growth inhibition zone present) and also included the concentrations of the drugs where sensitivity was observed and the maximum concentration used where resistance was observed. The actual quantitative MIC data for FK506 and APX879 in these fungi are included in the Table S4.

Strain	FK506 ($\mu\text{g/ml}$)	APX879 ($\mu\text{g/ml}$)
Aspergillus fumigatus ¹		
Wild type (AF293)	S (0.0156)	S (0.5)
Wild type (CEA10)	S (0.0312)	S (1.0)
Δfkbp12 (akuB ^{KU80})	R (>5)	R (>8)
Cryptococcus neoformans *		
Wild type (H99)	S (0.06)	S (1.0)
frr1 Δ (MCC1)	R (>25)	R (>25)
Wild type (JEC21)	S (0.06)	S (1.0)
frr1-1 (C20F1)	R (>25)	R (>25)
frr1-2 (C20F2)	R (>25)	R (>25)
CNB1-1 (C21F2)	R (>25)	R (>25)
frr1-3 (C21F3)	R (>25)	R (>25)
Candida albicans *		
Wild type (SC5314)	S (0.06)	S (8.0)
CNB1-1/CNB1 (YAG237)	R (>25)	R (>100)
rbp1 Δ / rbp1 Δ (YAG171)	R (>25)	R (>100)
Mucor circinelloides f. lusitanicus		
Wild type (R7B)	S (0.125)	S (4.0)
fkba Δ (RBM1) ¹	R (>2)	R (>32)
fkba-1 (SCV33)	R (>25)	R (>25)
fkba-2 (SCV41)	R (>25)	R (>25)
CNBR-1 (MSL11)	R (>25)	R (>25)

*Disk diffusion halo assay was used to detect sensitivity or resistance.¹RPMI cultures were used for testing drug sensitivity. S-Sensitive (growth inhibition zone present); R-Resistant (no zone of inhibition present)

REVIEWERS' COMMENTS:

Reviewer #1 (Remarks to the Author):

I have read the revised manuscript carefully, along with the responses provided by the authors to the comments of all the reviewers.

I am convinced that the authors have done a commendable job in thoroughly addressing all the raised points.

Further, they have now included new "in vivo" immunosuppressive and efficacy data of APX879 compared to FK506 in a murine model as well as the results of checkerboard assays for APX879 and FK506 in combination with clinically relevant antifungals against differential fungal species have been now included.

Overall, this is a very nice contribution that I am sure will be of interest to a wide range of medicinal chemists.

I highly recommend publishing this manuscript as it is in Nature Communications.

Doriano LAMBA, IC-CNR, Trieste (Italy)

Reviewer #2 (Remarks to the Author):

The authors did a very nice job in answering my comments, by either clarifying the issues I raised or/and by performing additional experimentations. I have no further comments.

Nature Communications: Reviewers' comments for revised version

Point by Point Response

Reviewer #1: (Remarks to the Author)

I have read the revised manuscript carefully, along with the responses provided by the authors to the comments of all the reviewers.

I am convinced that the authors have done a commendable job in thoroughly addressing all the raised points.

Further, they have now included new "in vivo" immunosuppressive and efficacy data of APX879 compared to FK506 in a murine model as well as the results of checkerboard assays for APX879 and FK506 in combination with clinically relevant antifungals against differential fungal species have been now included.

Overall, this is a very nice contribution that I am sure will be of interest to a wide range of medicinal chemists.

I highly recommend publishing this manuscript as it is in Nature Communications.

Doriano LAMBA, IC-CNR, Trieste (Italy)

Reviewer #2: (Remarks to the Author)

The authors did a very nice job in answering my comments, by either clarifying the issues I raised or/and by performing additional experimentations. I have no further comments.

Response: We appreciate the positive comments from both the reviewers and we are thankful for their comments that have significantly improved the overall manuscript.